# KUMA: A Novel Framework with Koopman Separation and Efficient Multilevel Extraction in Time Series Forecasting

Sijie Xiong [1]   Cheng Tang [1][2]   Atsushi Shimada [1]

## Abstract

Time series forecasting plays a crucial role in a wide range of real-world applications and has become increasingly complex with the growth of multivariate dimensions and extended historical observations, leading to the prosperity of deep forecasting models. Previous models are hindered by three major challenges: high computational complexity, inefficient token utilization caused by redundancy and scarcity, and temporal distribution shifts resulting from non-stationary dynamics. Inspired by Koopman theory and the success of multilevel encoder–decoder architectures with skip connections, we design an input-dependent **K**oopman module to decompose time series into Koopman dynamics and residual dynamics. Building upon this formulation, we propose a **U**-shaped **M**ultilevel **A**ttention module (UMA) that integrates element-wise attention filtering and linear attention, giving rise to KUMA. The input-dependent Koopman operator mitigates the issue of operator mixture and alleviates temporal distribution shifts, while UMA achieves a favorable balance between token redundancy and token scarcity with acceptable computational efficiency. Comprehensive evaluations across 12 benchmark datasets demonstrate that KUMA achieves superior performance compared to existing excellent approaches.

## 1. Introduction

Time series forecasting (TSF), which focuses on analyzing historical data to explore underlying patterns and predict future trends, has become essential in real-world applications, including financial assessments, weather forecasting, energy scheduling, and traffic flow management (Xiong et al., 2025b; Zhang et al., 2025; Chu et al., 2025). With the incorporation of more variates and the expanding availability of observations, deep learning approaches exhibit superior performance and promote the prosperity of deep forecasting models (Liu et al., 2023; Xiong et al., 2025a).

In the early stages of development, Temporal Convolutional Networks (TCNs) (Wu et al., 2023) and Recurrent Neural Networks (RNNs) (Hochreiter & Schmidhuber, 1997) employ convolutional kernels and recurrent architectures to model temporal dependencies (TDs), thereby improving prediction accuracy. However, the neglect of variate correlations (VCs) severely constrains the superiority of TCNs and RNNs (Liu et al., 2023; Ren et al., 2024). Subsequently, Transformer (Vaswani et al., 2017) endowed with attention mechanisms demonstrates an inherent advantage in modeling both TDs and VCs. Inspired by this advancement, numerous Transformer-based variants (e,g., iTransformer (Liu et al., 2024), PatchTST (Nie et al., 2023)) have been developed and have obtained great achievements in TSF (Wang et al., 2025b; Xiong et al., 2026a). Nevertheless, three inherent challenges remain in Transformer-based models: **(1) High Computational Complexity** The computational complexity reaches a quadratic level at $\mathcal{O}(L^2)$ and poses a significant challenge for long-distance and rich-variate forecasting. **(2) Token Redundancy** Global attention improves forecasting accuracy but introduces redundant tokens. (Bian et al., 2021; Xiong et al., 2025c). This occurrence leads to performance degradation on low-variate-density (LVD) datasets, while favoring Transformer-based models on high-variate-density (HVD) TSF scenarios (Xiong et al., 2025b). **(3) Temporal Distribution Shifts** A recognized assumption among them is that samples from both history and future are independent and identically distributed (Goodfellow et al., 2016; Liu et al., 2023). The world is inherently non-stationary and changes over time. The distribution of time series dynamically changes accordingly, generating temporal distribution shifts (TDS) (Zhang et al., 2024).

To reduce computational complexity, Linear-based approaches such as RLinear and DLinear have been proposed,

[1]Graduate School of Information Science and Electrical Engineering, Kyushu University, Fukuoka, Japan. [2]Department of Electrical and Mechanical Engineering, Nagoya Institute of Technology, Nagoya, Japan. Correspondence to: Sijie Xiong <sijiexiongkyushu@gmail.com>, Cheng Tang <tang.cheng@nitech.ac.jp>.

*Proceedings of the 43rd International Conference on Machine Learning*, Seoul, South Korea. PMLR 306, 2026. Copyright 2026 by the author(s).

dependent primarily on vanilla linear layers (Zeng et al., 2023). Nonetheless, as indicated by (Zeng et al., 2023), although Linear-based models make significant dumps in computational consumptions and achieve accuracy improvements in LVD, they tend to perform suboptimal under more limited or complex scenarios like HVD. Meanwhile, inspired by control engineering, (Gu & Dao, 2024) propose Mamba, a selective state space model with parallel computing. The parallel computing reduces the computational complexity to near-linear level and the selective mechanism extracts more nonlinear features compared to Linear-based models, achieving more balanced performance across HVD and LVD (Wang et al., 2025b). By contrast, restricted by the compact kernel of the system, Mamba is more insensitive to local details compared to Transformer-based models, suffering from **Token Scarcity** (Xiong et al., 2025b).

Alternatively, numerous studies (Katharopoulos et al., 2020; Han et al., 2024) adopt a more restrained strategy, which aims to maintain low computational complexity while avoiding token redundancy or scarcity. Katharopoulos et al. (2020) modify the attention computation order, making $\mathcal{O}(L^2)$ to $\mathcal{O}(L)$ and avoiding token redundancy. However, this process sacrifices token-to-token interaction, triggering token scarcity (Papa et al., 2024). Furthermore, Mamba-Inspired-Linear-Attention (MILA) (Han et al., 2024) demystifies Mamba and identifies crucial components to alleviate token redundancy while keeping nonlinear extraction ability. Like Mamba, MILA is still constrained by token scarcity. However, as demonstrated by U-Net (Ronneberger et al., 2015), U-Mixer (Ma et al., 2024), and U-KAN (Li et al., 2025), simple skip connections combined with a U-shaped architecture can provide a flexible and adjustable balance between redundancy and scarcity (Azad et al., 2024).

As for TDS, the inherent complexity of time series is overlooked by all these models. The heterogeneous components with distinct dynamics remain underexplored, which hinders forecasting accuracy and poses significant challenges for TSF (Liu et al., 2023). To tackle TDS, Koopa pioneers to adopt a mixture of Koopman operators to capture features under different dynamics, achieving remarkable prediction precisions (Liu et al., 2023). Koopa employs Fourier filters to disentangle time-variant and time-invariant series in frequency domain and then designs two kinds of Koopa blocks in temporal domain (Liu et al., 2023). There are no distinct separations and input-dependent adaptability from purely temporal domain for subordinate components. The forecasting performance can be affected by this alternation between both domains (Zhou et al., 2022).

In this work, we revisit and identify three fundamental challenges faced by TSF. To address these challenges in a unified manner, we propose KUMA, a framework that explicitly separates time series into Koopman dynamics and residual

dynamics as shown in Figure 1, and assigns each component a structurally designed processor. Specifically, we introduce an input-dependent **K**oopman processor, where the Koopman operator is conditioned on historical observations. This design avoids the implicit mixture of multiple operators under non-stationary inputs and serves as a structural fence that separates Koopman-governed dynamics from residual components, effectively reducing TDS. The residual dynamics, which exhibit hierarchical and multi-scale temporal patterns, are modeled by a **U**-shaped **M**ultilevel **A**ttention (UMA) processor. Rather than treating residuals as homogeneous noise, UMA adopts a multilevel encoder–decoder structure with skip connections to progressively refine residual representations. Its modular design integrates element-wise filtering with linear attention, achieving a favorable balance between modeling capacity and computational efficiency. Finally, KUMA fuses the Koopman and residual dynamics into a unified prediction, forming an efficient solution to TSF under real-world non-stationarity. Our contributions are summarized as follows:

- We identify three fundamental challenges in TSF and propose an **input-dependent Koopman processor** that adapts operators to input dynamics, alleviating TDS.

- We introduce a **multilevel residual modeling mechanism** that captures hierarchical structures beyond Koopman dynamics, instantiated by a simple yet effective UMA processor with flexible and adjustable balance in token redundancy and scarcity.

- We present **KUMA**, a unified framework that mechanistically separates and fuses Koopman and residual dynamics. Extensive experiments on 12 real-world datasets demonstrate its superior performance and competitive efficiency compared to excellent baselines.

## 2. Preliminary

### 2.1. Time Series Forecasting

In **Time Series Forecasting (TSF)**, the objective is to predict future states $Y \in \mathbb{R}^{N \times P}$ based on historical observations $X \in \mathbb{R}^{N \times L}$, where $N$ is the number of variates, $P$ is the forecasting horizon, and $L$ is the historical horizon. In essence, TSF can be viewed as learning a mapping function $f : \mathbb{R}^{N \times L} \to \mathbb{R}^{N \times P}$ (Xiong et al., 2026b).

### 2.2. Appropriate Koopman Operator

As in (Azencot et al., 2020), the time series stated in Section 2.1 can be described by a discrete-time evolution function as:

$$x_{t+1} = K(x_t), \tag{1}$$

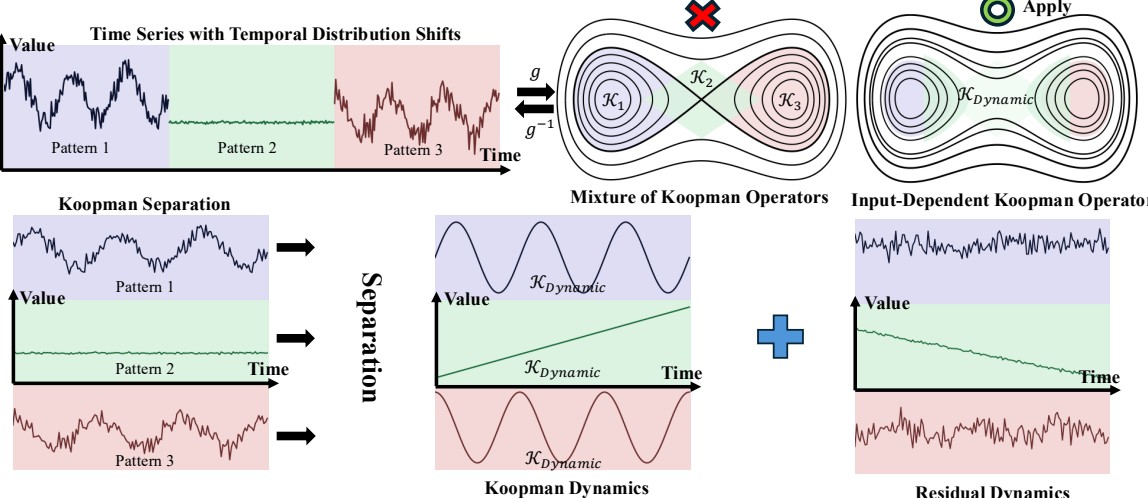

*Figure 1.* The overall illustration of input-dependent Koopman operator, Koopman dynamics, and residual dynamics. $g$ is the Koopman observation function denoted in Eq. (2). The Koopman operator separates the two kinds of dynamics. Since no mixture of Koopman operators is employed, it is essential to tackle the residual dynamics which contain abrupt change information among different patterns.

where $K(x_t)$ updates the state from discrete time $t$ to $t+1$ on an infinite dimensional manifold $\mathcal{X} \in \mathbb{R}^d$ and $d$ denotes the dimension of embedded $X \in \mathbb{R}^{C \times L} \xrightarrow{embed} X_{embed} \in \mathbb{R}^{C \times d}$. Koopman theory (Koopman, 1931) suggests that the nonlinear dynamics, Eq. (1), can be transformed into a Koopman space where the evolution becomes linear. Formally, for Eq. (1), there exists a linear finite dimensional Koopman operator $\mathcal{K} : \mathcal{G}(\mathcal{X}) \rightarrow \mathcal{G}(\mathcal{X})$ so that:

$$\mathcal{K}g(x_t) = g(K(x_t)) = g(x_{t+1}), \qquad (2)$$

where $g \in \mathcal{G}$ denotes an observation function in observation space $\mathcal{G}$, which transforms inputs into Koopman space.

In Koopman space, the dynamic, Eq. (2), is advanced forward by a combination of infinite-dimensional linear Koopman operators (Zhang et al., 2024). Time series under different distributions are treated by different Koopman operators. Therefore, a single-step time series forecasting can be concluded as:

$$x_{t+1} = \kappa(g(x_{t+1})) = \kappa(\mathcal{K}g(x_t)), \qquad (3)$$

where $\kappa$ is a function that reconstructs the original data from its observation space.

Naturally, the identification of an appropriate Koopman operator $\mathcal{K}$ constitutes the central challenge of applications of Koopman theory in TSF. A simple yet guiding shortcut for obtaining proper Koopman operators can provide meaningful insights and practical benefits for TSF.

## 3. Methodology

The design of KUMA adheres to the three issues discussed in Section 1. As illustrated in Figure 2, KUMA has two coordinated components: Koopman dynamic module (KDM) and U-shaped multilevel attention (UMA) module. KD is input-dependent and captures features relevant to temporal distribution shifts (TDS), which can naturally act as a dynamic fence and separate the embedded input as: Koopman dynamics and residual dynamics. Without a mixture of Koopman operators demonstrated in Figure 1, it is necessary to store the abrupt change information and local details in the residual dynamics (Appendix: C). Residual dynamics are further processed by UMA, which is a U-Shaped encoder-decoder structure with gated skip connections and flexible to acquire features at different granularity. Moreover, the multilayer structure benefits the balance between token redundancy and scarcity. The subordinate processor employed by UMA is Linear Attention (LA), leading to computational efficiency. We next provide a detailed description of each component.

All detailed algorithms can be found in Appendix: D.1.

### 3.1. KDM: Input-Dependent Koopman Separation

The complex temporal patterns drive mixtures of Koopman operators in current TDS reduction schemes as demonstrated in Section 2.2. Endless patterns require infinite operators. Herein, we design an input-dependent Koopman dynamic module to reduce TDS within a unified algorithm, which **is not a strict Koopman operator but is the inspired one**.

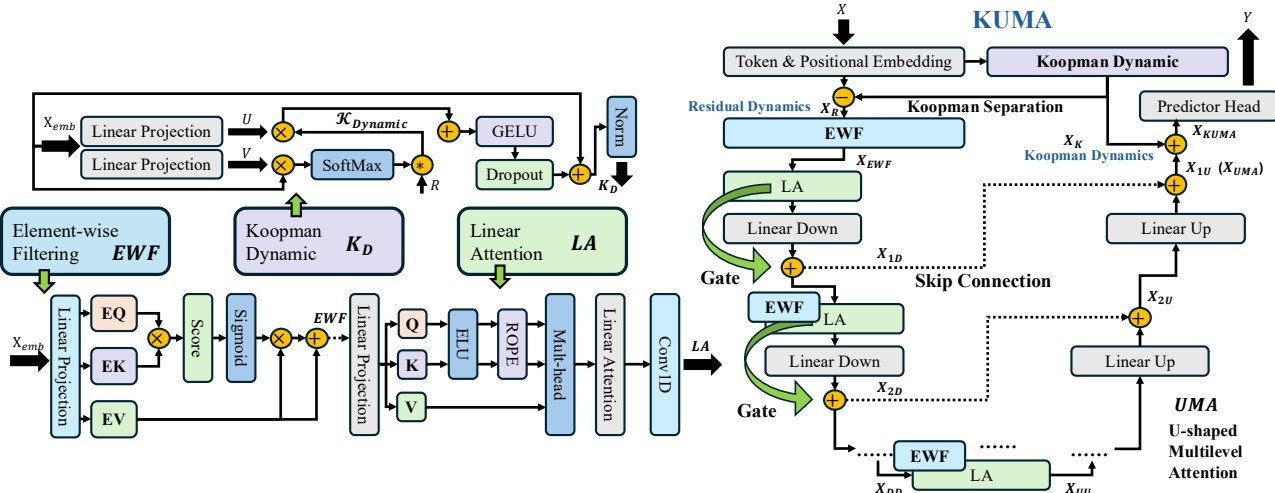

*Figure 2.* The overall architecture of KUMA (right) and its components (left): Koopman Dynamic module (KDM), element-wise filtering (EWF), and linear attention (LA). Gate contains a down-sampling and an sigmoid activation. ROPE represents rotary position embedding.

After a linear tokenization and a positional embedding, the historical sequence $X \in \mathbb{R}^{B \times N \times L}$ turns to be $X_{emb} \in \mathbb{R}^{B \times N \times D}$, with $D$ denoting the hidden dimension. To be input-dependent, $X_{emb}$ experiences two linear projections and differentiates into $U, V \in \mathbb{R}^{B \times N \times D \times R}$ so that $R$ represents the dimension of the unified Koopman space. Then, the Koopman operator is realized by:

$$\mathcal{K}_{Dynamic} = SoftMax(V \times X_{emb}) * R, \qquad (4)$$

where $\mathcal{K}_{Dynamic}$ is input-dependent, $SoftMax(\cdot)$ is an activation function in Koopman space, $R \in \mathbb{R}^{R \times R}$ is a learnable variate that boosts the Koopman space. $\times$ represents element multiplications and $*$ is matrix multiplications. The Koopman dynamics $K_D \in \mathbb{R}^{B \times N \times D}$ is estimated as:

$$K_D = \mathrm{Norm}(\mathrm{Dropout}(\mathrm{GELU}(\mathcal{K}_{Dynamic} \times U))), \quad (5)$$

where $\mathrm{GELU}(\cdot)$ is an activation function, $\mathrm{Dropout}(\cdot)$ has a dropout rate at $0.05$, $\mathrm{Norm}(\cdot)$ means normalization.

As presented in Figure 1, the drawback is that KDM cannot precisely react to local details and abrupt changes in the asymptotic regions marked in light green. However, these dynamics are of great importance to forecasting accuracy and they should be recovered. We further conduct the Koopman separation to obtain Koopman dynamics $X_K$ and residual dynamics $X_R \in \mathbb{R}^{B \times N \times D}$ as:

$$\textbf{Separation:} \ X_K = K_D; X_R = X_{emb} - X_K. \quad (6)$$

### 3.2. UMA: Balanced Token Redundancy and Scarcity

**U-shaped Multilevel Attention (UMA)** UMA is designed to tackle the residual dynamics as separated in Eq. (6) and

balance token issues with acceptable computational efficiency. Adhering to these standards, UMA is composed of the element-wise filtering (EWF), encoders consisting of linear attention and linear down-sampling projection with a gate, a linear projection bottom, decoders formed by linear up-sampling projection, skip connections, and a predictor head. The multilevel structure ensures the flexibility of token redundancy and scarcity. While most of the components are composed by linear projections, the computational efficiency can be preserved.

**Element-Wise Filtering** For multilayer architectures, coarse inputs tend to introduce a Matthew effect[1], in which errors and irrelevant features are progressively amplified in deeper layers. To capture both TD and VC, while avoiding quadratic complexity, we design an element-wise filtering mechanism (EWF). Similar to attention mechanisms, we initially conduct linear projections to form $EQ, EK, EV \in \mathbb{R}^{B \times N \times D}$. Then element multiplication is applied to $EQ, EK$ to get the score. After a sigmoid activation, the Weighting of filtration can be determined. Finally, the filtered features $X_{EWF}$ can be computed as:

$$X_{EWF} = \mathrm{Dropout}(Weighting \times EV + EV). \quad (7)$$

Given refined features exist in all layers, we apply EWF at the entrance of each layer before any linear attention and do not implement it only once.

**Linear Attention** Vanilla multi-head attention increases the computational complexity given multiple levels and is

---

[1]Herein, we use Matthew effect as an intuitive motivation to describe the tendency that stronger informative signals may be further emphasized while weaker noisy components are suppressed.

easy to be affected by overfit caused by token redundancy. Inspired by (Liu et al., 2023), we remain the key components of Mamba equivalent to fast linear attention (Katharopoulos et al., 2020) and design the Linear Attention module (LA). Considering we conduct down-sampling to a half layer by layer, token or positional embeddings are destroyed (Han et al., 2024; Su et al., 2024). We employ Rotary Positional Embedding (ROPE) proposed in (Su et al., 2024) to exert re-embedding at each layer. The re-embedded features are further split into multiple heads and further conduct fast linear attentions. In the tail, we employ a convolution in 1-D dimension to integrate all extractions and output $LA \in \mathbb{R}^{B \times N \times d}$ with $d$ is equal to the last dimension of the input of the LA module. The entire process of LA module is at nearly linear level, which is deemed as computational efficiency and is further leveraged as the subordinate processor.

**Encoder and Bottom** In a single encoder, an LA module is employed as the processor to capture fine features. Then we employ linear projection to down-sample the last dimension to the half progressively. Moreover, to avoid token scarcity, we add a gate to control the fused contents. The down-sampling process can be inferred as the instance that: $X_{1D} \in \mathbb{R}^{B \times N \times D}$ after a layer becomes $X_{2D} \in \mathbb{R}^{B \times N \times D//2}$. This process continues until $X_{DD} \in \mathbb{R}^{B \times N \times D//(2^k)}$ with $k$ layers. The bottom is a linear attention that maps $X_{DD}$ to $X_{UU} \in \mathbb{R}^{B \times N \times D//(2^k)}$.

**Decoder and Skip Connection** In a single decoder, we employ merely a linear projection at each layer to recover the down-sampled dimension progressively back to $D$. Following every up-sampling, the recovered features take in the features from the corresponding encoder to prevent distortion via a naive skip connection: addition operation. Finally, one more linear projection is adopted as the predictor head to generate the prediction $Y \in \mathbb{R}^{B \times N \times P}$.

# 4. Experiments

In this section, to clarify fundamentals, extensive experiments and analyses are conducted on 12 recognized datasets including both LVD and HVD datasets from various domains. Long-term and short-term tasks are implemented to evaluate the performance of KUMA and its effectiveness on the three specific challenges.

**Datasets** The project is validated by 12 publicly available datasets and their specifications are presented in Appendix: E.1. In terms of the numbers of variates, we can divide them into 2 major categories: **Low-Variate-Density (LVD)** includes Exchange (Kitaev et al., 2020), Weather (Kitaev et al., 2020), ETTm1, ETTm2, ETTh1, ETTh2 (Kitaev et al., 2020); **High-Variate Density (HVD)** includes Electricity (Kitaev et al., 2020), Solar-Energy (Lai et al., 2018), PEMS03, PEMS04, PEMS07, PEMS08 (Chen et al., 2023).

For more details, refer to Appendix: E.1.

**Baselines** To confirm that achievements are attained by KUMA in the three issues, we select representatives from each mainstream of models. Transformer-based models: iTransformer (Liu et al., 2024), PatchTST (Nie et al., 2023), Crossformer (Zhang & Yan, 2023), FEDformer (Zhou et al., 2022), Autoformer (Wu et al., 2021). Linear-based models: RLinear (Zeng et al., 2023), DLinear (Zeng et al., 2023), TiDE (Das et al., 2023). TCN-based models: TimesNet (Wu et al., 2023). Mamba-based models: S-Mamba (Wang et al., 2025b). Koopman-based models: Koopa (Liu et al., 2023). Multilevel-based models: U-Mixer (Ma et al., 2024), TimeMixer (Wang et al., 2024). Mechanisms: SparseTSF (Lin et al., 2026), WaveRoRA (Liang et al., 2026). For details of these baselines and the selection motivations, refer to Appendix: E.2.

**Experimental Implementation** In this work, Mean Square Error (MSE) and Mean Absolute Error (MAE) are selected as the evaluation metrics. In addition, MSE acts as the loss function. Adam is the optimizer. The learning rate is set within $1e-3$ to $1e-5$, and progressively decays at $0.5$. Random seeds are set within 2021 to 2025. The number of heads is set at 4. All experiments are built upon PyTorch and NVIDIA RTX A100 GPU with 80 GB of memory. A table that concludes hyper-parameters and details on implementations are presented on GitHub.

# 5. Performance Analysis

**Overall Accuracy Analysis** Table 1 demonstrates all averaged results of four prediction horizons across 12 datasets and 15 baselines. Lower MSE and MAE indicate better forecasting ability. Compared with strong benchmarking models, KUMA achieves the best performance in 14 out of 24 situations and attains suboptimal results in 5 cases. This demonstrates that KUMA has consolidated superiority over other models and is not affected by random variance.

**Comparison: Koopman** Koopa achieves the best in ETTm1, which has 7 variates but severely deteriorates in 4 PEMS datasets whose numbers of variates go beyond 100. However, KUMA achieves excellent performance in both LVD and HVD. This supports our motivation that an increasing number of variates requires a large number of Koopman operators and validates the necessity of the input-dependent Koopman module. The extraordinary achievements in all datasets also recognize that KUMA is a unified model.

**Comparison: Multilevel** As presented in Table 1, U-Mixer and TimeMixer show competitiveness to KUMA in LVD datasets such as Exchange, Weather, ETT. However, their performances in HVD are similar to those of RLinear and DLinear. Notably, PEMS results demonstrate that U-Mixer and TimeMixer are ranked between RLinear and DLinear

*Table 1.* The results of all main experiments. All results are obtained with historical observation length $L = 12$ for PEMS datasets and $L = 96$ for the other datasets. The prediction windows are set within $\{12, 24, 48, 96\}$ for PEMS datasets and $\{96, 192, 336, 720\}$ for the rest. All the results disclosed herein are averaged and complete results can be found in Appendix: E.4.

| Dataset | KUMA | | Koopa | | U-Mixer | | WaveRoRA | | TimeMixer | | SparseTSF | | S-Mamba | | iTransformer | | RLinear | | PatchTST | | Crossformer | | TiDE | | TimesNet | | DLinear | | FEDformer | | Autoformer | |
|---|---|---|---|---|---|---|---|---|---|---|---|---|---|---|---|---|---|---|---|---|---|---|---|---|---|---|---|---|---|---|---|---|
| | MSE | MAE | MSE | MAE | MSE | MAE | MSE | MAE | MSE | MAE | MSE | MAE | MSE | MAE | MSE | MAE | MSE | MAE | MSE | MAE | MSE | MAE | MSE | MAE | MSE | MAE | MSE | MAE | MSE | MAE | MSE | MAE |
| Exchange | 0.355 | *0.400* | 0.363 | 0.407 | *0.349* | **0.399** | 0.369 | 0.412 | **0.332** | 0.426 | 0.354 | 0.406 | 0.367 | 0.408 | 0.360 | 0.403 | 0.378 | 0.417 | 0.367 | 0.404 | 0.940 | 0.707 | 0.370 | 0.413 | 0.416 | 0.443 | 0.354 | 0.414 | 0.519 | 0.429 | 0.613 | 0.539 |
| Weather | **0.245** | **0.273** | 0.248 | 0.281 | 0.248 | *0.276* | 0.247 | 0.283 | **0.238** | 0.281 | 0.291 | 0.303 | 0.251 | *0.276* | 0.258 | 0.278 | 0.272 | 0.291 | 0.259 | 0.281 | 0.259 | 0.315 | 0.271 | 0.320 | 0.259 | 0.287 | 0.265 | 0.317 | 0.309 | 0.360 | 0.338 | 0.382 |
| Solar-Energy | *0.235* | 0.267 | 0.245 | 0.281 | 0.244 | 0.275 | **0.235** | **0.263** | 0.244 | 0.296 | 0.282 | 0.316 | 0.240 | 0.273 | **0.233** | **0.262** | 0.369 | 0.356 | 0.270 | 0.307 | 0.641 | 0.639 | 0.347 | 0.417 | 0.301 | 0.319 | 0.330 | 0.401 | 0.291 | 0.381 | 0.885 | 0.711 |
| ECL | **0.167** | **0.263** | 0.172 | 0.271 | 0.178 | 0.270 | **0.168** | **0.263** | 0.172 | 0.274 | 0.223 | 0.288 | 0.170 | *0.265* | 0.178 | 0.270 | 0.219 | 0.298 | 0.205 | 0.290 | 0.244 | 0.334 | 0.251 | 0.344 | 0.192 | 0.295 | 0.212 | 0.300 | 0.214 | 0.327 | 0.227 | 0.338 |
| ETTm1 | 0.387 | *0.398* | **0.376** | **0.395** | 0.388 | *0.395* | 0.399 | 0.399 | *0.381* | 0.414 | 0.415 | 0.401 | 0.398 | 0.405 | 0.407 | 0.410 | 0.414 | 0.407 | 0.387 | 0.400 | 0.513 | 0.496 | 0.419 | 0.419 | 0.400 | 0.406 | 0.403 | 0.407 | 0.448 | 0.452 | 0.588 | 0.517 |
| ETTm2 | **0.277** | **0.322** | 0.283 | 0.328 | 0.301 | 0.337 | 0.282 | 0.329 | 0.283 | 0.347 | 0.300 | 0.331 | 0.288 | 0.332 | 0.288 | 0.332 | 0.286 | 0.327 | *0.281* | *0.326* | 0.757 | 0.610 | 0.358 | 0.404 | 0.291 | 0.333 | 0.350 | 0.401 | 0.305 | 0.349 | 0.327 | 0.371 |
| ETTh1 | *0.438* | 0.438 | 0.445 | 0.446 | 0.448 | 0.440 | 0.454 | 0.474 | 0.444 | **0.433** | | | 0.455 | 0.450 | 0.454 | 0.447 | 0.446 | *0.434* | 0.469 | 0.454 | 0.529 | 0.522 | 0.541 | 0.507 | 0.458 | 0.450 | 0.456 | 0.452 | *0.440* | 0.460 | 0.496 | 0.487 |
| ETTh2 | **0.374** | 0.401 | 0.380 | 0.405 | 0.383 | 0.402 | **0.365** | **0.397** | 0.379 | 0.426 | 0.391 | 0.408 | 0.381 | 0.405 | 0.383 | 0.407 | *0.374* | *0.398* | 0.387 | 0.407 | 0.942 | 0.684 | 0.611 | 0.550 | 0.414 | 0.427 | 0.559 | 0.515 | 0.437 | 0.449 | 0.450 | 0.459 |
| PEMS03 | **0.109** | **0.215** | 0.136 | 0.243 | 0.331 | 0.395 | 0.327 | 0.394 | 0.186 | 0.469 | 0.206 | 0.323 | 0.122 | 0.228 | *0.113* | *0.221* | 0.495 | 0.472 | 0.180 | 0.291 | 0.169 | 0.281 | 0.326 | 0.419 | 0.147 | 0.248 | 0.278 | 0.375 | 0.213 | 0.327 | 0.667 | 0.601 |
| PEMS04 | **0.085** | **0.192** | 0.116 | 0.228 | 0.317 | 0.393 | 0.327 | 0.395 | 0.280 | 0.366 | 0.218 | 0.312 | *0.103* | *0.211* | 0.111 | 0.221 | 0.526 | 0.491 | 0.195 | 0.307 | 0.209 | 0.314 | 0.353 | 0.437 | 0.129 | 0.241 | 0.295 | 0.388 | 0.231 | 0.337 | 0.610 | 0.590 |
| PEMS07 | **0.075** | **0.171** | 0.220 | 0.318 | 0.298 | 0.370 | 0.301 | 0.367 | 0.123 | 0.378 | 0.153 | 0.263 | *0.089* | *0.188* | 0.101 | 0.204 | 0.504 | 0.478 | 0.211 | 0.303 | 0.235 | 0.315 | 0.380 | 0.440 | 0.124 | 0.225 | 0.329 | 0.395 | 0.165 | 0.283 | 0.367 | 0.451 |
| PEMS08 | **0.129** | **0.215** | *0.146* | *0.221* | 0.335 | 0.403 | 0.337 | 0.411 | 0.330 | 0.400 | 0.266 | 0.337 | 0.148 | 0.224 | 0.150 | 0.226 | 0.529 | 0.487 | 0.280 | 0.321 | 0.268 | 0.307 | 0.441 | 0.464 | 0.193 | 0.271 | 0.379 | 0.416 | 0.286 | 0.358 | 0.814 | 0.659 |

**Color code: red** = best, *blue* = second best.

and indicate that the multilevel structure can help to adjust token issues. KUMA's superiority over U-Mixer and TimeMixer can be attributed to the adoption of linear attention mechanisms combined with element-wise filtering. As a result, token scarcity is significantly alleviated, leading to enhanced performance.

**Comparison: Mamba** The linear attention design is inspired by the similarity between Mamba and the attention mechanism. By removal of redundant components owned by Mamba, the linear attention block greatly reduces token redundancy. This is crucial in multilevel structure. Otherwise, accumulative redundancy contaminates fine features layer by layer. The simplicity of linear attention rather than Mamba is rationale.

**Comparison: Transformer** KUMA performs stronger than Transformer-based models. Although competitive models such as iTransformer and PatchTST reduce token redundancy through variate-wise modeling and patch-based representations, their dependence on global attention remains a limiting factor. In particular, improvements in VC and local TD modeling are insufficient to fully mitigate the adverse effects introduced by global attention mechanisms. The element-wise filtering and the sparsity of linear attention are confirmed to be reasonable in TSF.

**Comparison: Mechanisms** WaveRoRA is competitive to KUMA in LVD cases and some HVD cases. However, KUMA beats WaveRoRA in PEMS datasets. This reveals that frequency-domain analysis can be beneficial in TSF, however, KUMA provides an effective alternative separation. Compared to SparseTSF, KUMA demonstrates superiority in most cases. This certifies that KUMA effectively alleviates token redundancy and scarcity.

**Trackability Analysis** The accuracy can be visualized by the tracking trajectory. We present trajectories from 11 datasets in Figure 3 with 200 steps in each chart. As indicated by HVD datasets: PEMS, Electricity, Solar-Energy, with sufficient variates, KUMA can capture both local details and global trends. However, in LVD datasets such as ETT and Weather datasets, KUMA can track well with the tendency, while it remains struggling to respond to local features with sufficient amplitude. This suggests that KUMA can be further improved to extract more features from limited data. In contrast, the superior trend-tracking capability underscores the role of the Koopman operator. As discussed in Figure 1, Koopman operators excel at extracting macro-level dynamics from complex time series. The performance confirms that our original inspiration is preserved.

**Computational Complexity Analysis** To validate whether KUMA solves the quadratic computational complexity, we collect latency (inference time) and memory allocated for all experiments and present the average values in Figure 4. As illustrated in Figure 4, KUMA lies in the lower-left region, demonstrating an advantageous accuracy and efficiency trade-off with both low inference latency and high predictive accuracy. In addition, the memory occupation indicated by the bubble size demonstrates that KUMA is ranked in the middle. Due to the multilevel structure, KUMA occupies more memory than S-Mamba, Linear-based models. However, the benefits are that KUMA captures better features, leading to reduced errors. Moreover, according to Appendix: D.2, in LVD datasets such as Exchange and ETT, KUMA's performance in latency and memory is similar to that of Koopa. However, with the number of variates growing, Koopa deteriorates. This confirms the assumption that more variates require more Kooopman operators and the necessity of the input-dependent design. Compared to U-Mixer, as the main pattern is captured by KDM and linear attentions have finer representations, $D$ is extended to a vast space, accelerating latency and reducing the computational consumptions. Overall, KUMA is recognized an efficient multilevel model, reaching great equilibrium among the three challenges.

# 6. Ablation Study

**Component Effectiveness Analysis** In this section, the effectiveness of each component is explored independently. We conduct six ablation experiments to identify the contribution of each component. The six ablation models include: (1) **UMA**: The KDM is removed from KUMA; (2)

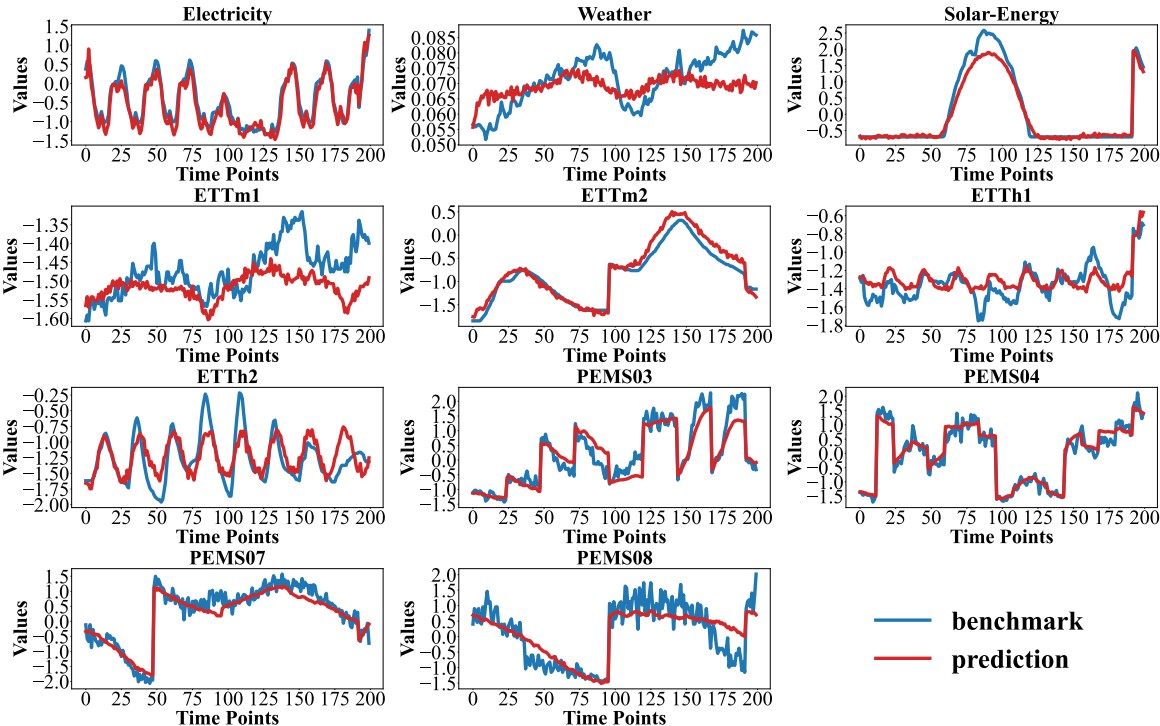

*Figure 3.* The tracking trajectories of KUMA across 11 representative datasets. In each subplot, 200 time steps are randomly selected. The more overlapping, the better forecasting capability.

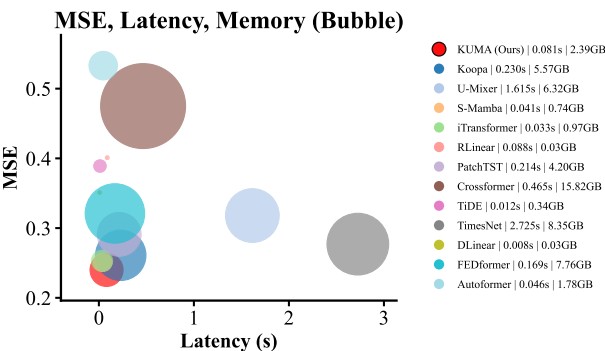

*Figure 4.* The averaged latency (*s*) and memory (*GB*) comparisons with baselines combined with averaged MSE and MAE.

**NA**: All linear attentions of KUMA are replaced by vanilla multi-head attention blocks with 4 heads; (3) **NoEWF**: All the EWF modules are deleted from KUMA; (4) **NoROPE**: The ROPE function in linear attentions are canceled; (5) **Mamba**: All linear attentions are replaced with Mamba blocks; (6) **NK**: We cancel the input-dependent mechanism of KDM and the whole Koopman space are learned during

processes. The results are visualized by averaged MSE in Figure 5 and complete results are presented in Appendix: E.5. The KDM contributes 3.4% enhancements in MSE. Further comparing UMA with NK, it can be found that the input-dependent mechanism gains in 1.6% (5% − 3.4%). In addition, EWF nearly contributes the same as KDM, which shows the importance of filtering and adheres to our original design. When compared to NA and Mamba, the vanilla attention generates a large number of redundant tokens deteriorating by 14.7% while Mamba is affected adversely by redundant elements dropping 4.4% in accuracy. This demonstrates the superiority to adopt simplified linear attentions in TSF. By contrast, ROPE has the most trivial contribution at 1.1%. Without RoPE, NoROPE fails to maintain stable positional representations on LVD datasets, leading to degraded predictive accuracy.

**Layer Effect Analysis** In addition to components, the multilevel structure can also influence performance of KUMA. Therefore, we test 1 to 4 layer(s). The averaged MSE and MAE for KUMA with different layers are presented in Table 2. The best results are marked in bold and red. It is evident that KUMA with 2 layers occupies the best in most cases. For a better generalization, throughout the work, we set KUMA with 2 layers. Besides, we find the variations of

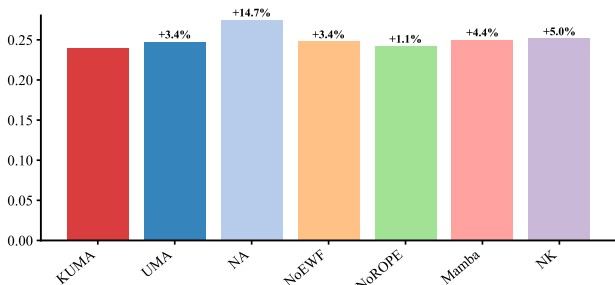

*Figure 5.* The averaged MSE and change ratios of KUMA and its variants. A positive movement indicates deteriorated performance.

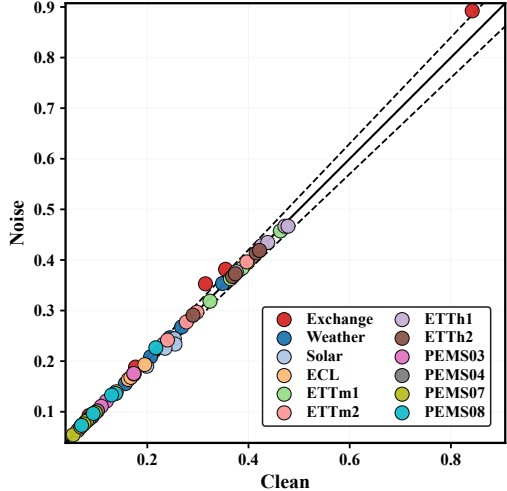

*Figure 6.* Scatter plots comparing clean (noise-free) and noisy performance of KUMA. Most points lie within the $\pm5\%$ band, indicating strong robustness to noise.

layers do not introduce huge differences. This is rationale, as in our original construction, the UMA part is employed to capture valuable features and the alternations from the residual dynamics. The principle dynamics have already captured by KDM. However, the variations matter. For example, for LVD datasets like Exhcange, ETT, and Weather, 2-level structure is sufficient, while for HVD datasets like PEMS, 4-level structure can serve better. Overall, these results show that the multilevel structure is flexible to adjust token redundancy and scarcity.

*Table 2.* Averaged MSE and MAE across layers (lower is better).

| Dataset | Layer 1 | | Layer 2 | | Layer 3 | | Layer 4 | |
|---|---|---|---|---|---|---|---|---|
| | MSE | MAE | MSE | MAE | MSE | MAE | MSE | MAE |
| Exchange | 0.358 | 0.402 | **0.355** | **0.400** | 0.359 | 0.405 | 0.359 | 0.403 |
| Weather | 0.249 | 0.275 | **0.245** | **0.273** | **0.245** | **0.273** | 0.247 | 0.275 |
| Solar-Energy | **0.234** | 0.268 | 0.235 | **0.267** | **0.234** | 0.268 | 0.236 | 0.269 |
| ECL | 0.170 | 0.264 | **0.167** | **0.263** | 0.170 | 0.264 | 0.174 | 0.266 |
| ETTm1 | **0.386** | **0.397** | 0.387 | 0.398 | 0.387 | **0.397** | 0.387 | 0.398 |
| ETTm2 | **0.277** | **0.322** | **0.277** | **0.322** | 0.278 | **0.322** | 0.278 | 0.323 |
| ETTh1 | 0.444 | 0.441 | **0.438** | **0.438** | 0.439 | 0.439 | 0.446 | 0.441 |
| ETTh2 | 0.378 | 0.403 | **0.374** | **0.401** | 0.379 | 0.404 | 0.381 | 0.406 |
| PEMS03 | 0.110 | 0.216 | **0.109** | **0.215** | 0.109 | 0.215 | 0.109 | 0.215 |
| PEMS04 | 0.088 | 0.196 | 0.085 | 0.192 | 0.085 | 0.191 | **0.084** | **0.190** |
| PEMS07 | 0.077 | 0.174 | 0.075 | 0.171 | 0.075 | 0.170 | **0.074** | **0.169** |
| PEMS08 | **0.124** | 0.215 | 0.129 | 0.215 | **0.124** | **0.213** | 0.131 | 0.215 |

**Resistance to Noise** With Koopman operator and the multilevel structure, KUMA is robust to noise. First, the random noise under certain level does not affect the manifold of Koopman space. Second, with EWF and layer by layer filtering, the random noise can be mitigated during the process. To validate this motivation, we randomly select 5 out of 96 time points of the input (the disturbance strength is at $5.2\% = 5/96$) and add Gaussian noises to all variates. If the deterioration in MSE and MAE remains below $5\%$, KUMA can be considered robust to noise. As presented in Figure 6, most points of all situations are bounded by the $\pm5\%$ borders except for one point from Exchange. Overall, KUMA has significant robustness to random noise, ensuring its applicability and flexibility.

**Robustness: Random Variance** To ensure the superiority of KUMA does not come from random variance, all the results (**Final**) are averaged from 5 random seeds: 2021 to 2025. In addition, we select 10 to 14 as the control group to repeat implementations. The results are disclosed in Figure 7. In all datasets, the final reported results are nearly at the same level as those of random seeds from 10 to 14. Despite the fact that the superiority of KUMA is not generated by random variance, the reproducibility of KUMA can be ensured.

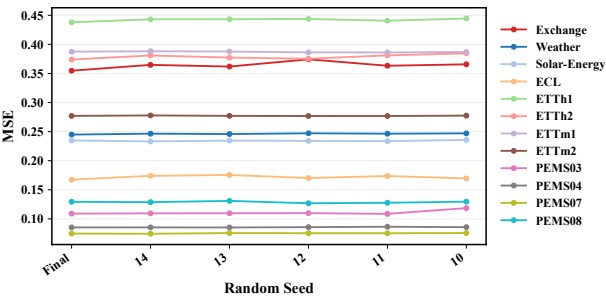

*Figure 7.* The averaged MSE of the final results and those of other random seeds. A horizontal line shows the reproducibility possibility in each datasets. All lines are nearly horizontal.

**Robustness: Confidence Level** As illustrated in Fig. 8, in most datasets and under different random seeds, KUMA controls the variations below 5%. It admits that in PEMS03, KUMA varies greatly under different seeds. Moreover,

MAE demonstrates more stable variations compared to MSE. Overall, according to total average results, KUMA performs consistently under different random seeds, consolidating the robustness of KUMA. A complete performance is presented in Table 10.

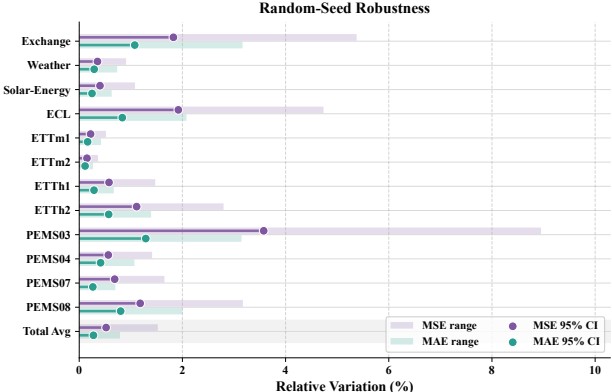

*Figure 8.* The illustration of the averaged relative variation generated from random seeds across datasets. The green and purple points are 95% CI values for MAE and MSE. The green and purple lines represent min-max range for MAE and MSE.

**Radar: Statistical Test** As presented in Fig. 9, KUMA occupies the largest area compared to all baselines. This shows that KUMA has superiority over other counterparts in terms of accuracy. Moreover, as presented in Table 11 and 12, all MSE results are significant and most cases are significant in MAE results. This further consolidates the superiority of KUMA over benchmarks. The details of Friedman nonparametric test and Wilcoxon test are presented in Table 11 and Table 12.

**Separation: UMAP & Silhouette** As visualized in Fig. 13, KUMA can collect data of one-class in a more clustered manner. This visualizes the separation function of KDM. Moreover, quantified by Silhouette scores, it is clear that the Koopman dynamics occupy the highest score followed by residual signals and original undistinguishable signals, which means better clustering. In general, KUMA adheres to our original intention of separating different signals.

**Separation: Component Identification** Fig. 14 shows that the Koopman component preserves the most salient features of the original embedding, whereas the residual signals exhibit patterns that are substantially different from the original signals. This observation supports our claim that KDM effectively extracts dominant patterns, while the residual signals retain complementary information, such as abrupt changes and local variations.

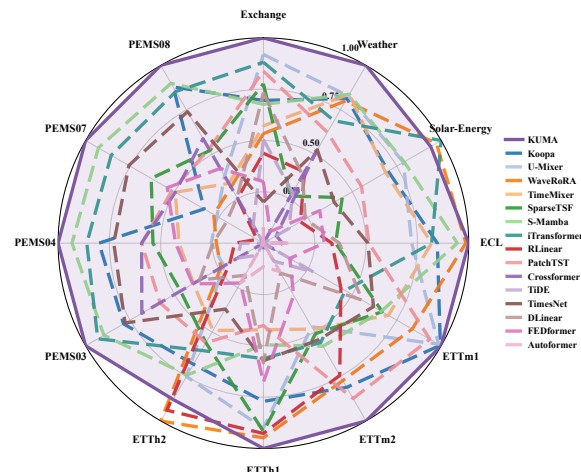

*Figure 9.* The radar chart of KUMA and baselines based on Friedman nonparametric test MSE results.

# 7. Conclusions

This work focuses on three specific challenges in time series forecasting: high computational complexity, token redundancy and scarcity, temporal distribution shifts (TDS) and proposes KUMA. By designing an input-dependent Koopman module, KUMA avoids mixtures of Koopman operators while reducing TDS. To capture local details and abrupt change information, we design the U-shaped multilevel linear attention (UMA). The element-wise filtering and the simplified linear attention blocks with strong representations preserve near-linear computational complexity. Moreover, the introduction of the multilevel backbone makes KUMA flexible in token redundancy and scarcity. Extensive main experiments demonstrate that KUMA overcomes the three challenges and has superiority over excellent benchmarks in accuracy with great computational efficiency. Further ablation studies reveal that each component design adheres to our original motivation and contributes to the success of KUMA. Each component contributes more than 3% efforts. Evidences show a two-layer KUMA possesses the most flexibility for numerous domains. The stressful noise test reveals that KUMA has excellent ability to resist random noise, while with numerous random seeds, the results remain consistent, ensuring the reproducibility. The statistical results from the t-test, Friedman nonparametric test, and Wilcoxon signed-rank test demonstrate that KUMA performs consistently across different random seeds, and that its superiority is statistically significant rather than occurring by chance. Overall, KUMA is a novel time series framework and has promising characteristics to be further explored. One potential direction is to equip KUMA with appropriate stimulus strengths.

## Acknowledgment

This work was supported by JSPS KAKENHI Grant Number JP22H00551 and JP25K21340, and JST CREST Grant Number JPMJCR22D1, Japan.

## Impact Statement

This paper presents work whose goal is to advance the field of machine learning. There are many potential societal consequences of our work, none of which we feel must be specifically highlighted here.

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

# A. Articulation

**Unified**: 'Unified' and its variants are employed in this work. Throughout the entire work, we define "Unified" as the consistent and excellent performance across both low-variate-density (LVD) and high-variate-density (HVD) datasets. A model can be unified only when it can achieve excellent performance in all datasets employed. For example, Linear-based models such as RLinear and DLinear are not unified as they underperform in HVD.

**U-shaped Multilevel Attention (UMA)**: This is inspired by the U-Net (Ronneberger et al., 2015) architecture, where skip connections are employed to facilitate information flow between the encoder and the decoder. There are two reasons that we do not employ U-Net to define this kind of structure. First, the input of U-Net is usually in four dimensions as [Batch Size, Channel, Height, Width] while most multivariate time series input is displayed as [Batch Size, Variate, Time]. We do not want to confuse readers with contradictory to common sense in U-Net. Second, the subordinate processors acting as encoders and decoders are linear attention blocks and linear projections, respectively. They are not fully convolutional blocks, while U-Net-based models frequently possess convolutions. We consider that UMA shares a similar form with U-Net but differs in substance.

**Temporal Dependency (TD) and Variate Correlation (VC)** TD refers to temporal relationships among time steps, while VC refers to the intersection between variates. TD and VC can be visualized by Figure 10.

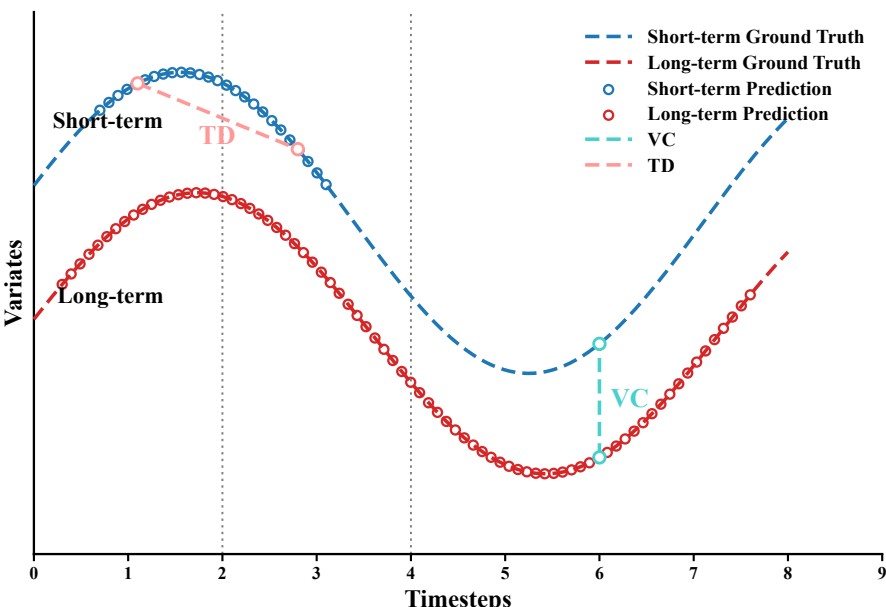

*Figure 10.* The visualization of temporal dependency (TD) and variate correlation (VC).

# B. Related Works

**Time Series Forecasting (TSF)** TSF predicts future patterns based on historical observations. The mainstream approaches can be divided into three categories: (1) statistical learning methods (SL); (2) machine learning methods (ML); (3) deep learning methods (DL) (Xiong et al., 2025c; Wu et al., 2025). SL-based methods such as ARIMA (Box & Pierce, 1970) are highly dependent on complex dynamics. The extension of lengths and the growth of variates can both introduce high-dimensional nonlinearity, which hinders forecasting. This drawback is overcome by the advancements in machine learning. ML-based models such as XGBoost (Chen, 2016) and LightGBM (Ke et al., 2017) gain popularity for handling nonlinear dynamics (Wu et al., 2025). However, ML-based methods can produce reliable forecasts even with limited data; nevertheless, their performance depends on extensive domain-expert feature engineering (Wu et al., 2025). With the explosion of observations across numerous domains, TD and VC become extremely complex and ML-based methods

struggle to model such complex multivariate time series forecasting tasks. This promotes the prosperity of DL-based models.

**Transformer** One representative of DL-based models is Transformer, which can capture TD and VC at the same time via the attention mechanism. Transformer has achieved outstanding achievements in natural language processing, computer vision, and other aspects (Han et al., 2023). Recently, numerous Transformer-based models have been proposed in TSF, including Informer (Zhou et al., 2021), iTransformer (Liu et al., 2024), PatchTST (Nie et al., 2023), Crossformer (Zhang & Yan, 2023), FEDformer (Zhou et al., 2022), Autoformer (Wu et al., 2021), TimeBridge (Liu et al., 2025), LightGTS (Wang et al., 2025a). Although they achieve improved accuracy through diverse attention mechanisms, these approaches incur quadratic computational complexity. *High computational complexity* becomes a heavy burden for Transformer-based models. Moreover, global attentions trigger *token redundancy* (Long et al., 2023; Feng & Zhang, 2023). (Xiong et al., 2025c; Zeng et al., 2023) argue that under certain scenarios, a simple linear projection also generates competitive performance.

**Linear and Multilevel** As alternatives to Transformer-based models, Linear-based models focus on combinations of linear projections but still achieve comparative performance in TSF. NLinear (Zeng et al., 2023), DLinear (Zeng et al., 2023), SparseTSF (Lin et al., 2026), UMixer (Ma et al., 2024), and AMD (Hu et al., 2025) validate that under simple scenarios with few variates, Linear-based models can outperform complex models. However, as the number of variates increases, the capacity of linear modules becomes insufficient to capture rich and expressive token representations (Xiong et al., 2025b; Cao et al., 2023). Accordingly, with *token scarcity*, the performance of linear-based models degrades. However, U-Mixer (Ma et al., 2024) and AMD (Hu et al., 2025) which have multilevel structures, demonstrate a more slight deterioration in intricate situations, revealing that multiple layers can alleviate such scarcity and indicating a promising scheme to adjust *token redundancy and token scarcity* among Transformer-based models and Linear-based approaches.

**Koopman** Additionally, the mentioned approaches frequently adopt statistical methods to reduce the non-stationary and do not pay sufficient attention to *temporal distribution shifts (TDS)* (Liu et al., 2023). Inspired by Koopman operator that can reduce TDS from manifolds (Koopman, 1931; Zhang et al., 2024), (Zhang et al., 2024; Liu et al., 2023) take advantage of mixtures of Koopman operators to solve TDS issues. However, modeling numerous patterns using a mixture of Koopman spaces significantly increases memory consumption. (Wu et al., 2025) validate this limitation and propose $K^2$VAE for long-horizon forecasting; nevertheless, its shared Koopman dynamics depend strongly on posterior Kalman inference and often represent averaged system dynamics rather than regime-specific behaviors.

## C. Koopman Necessitates U-shaped Multilevel Attention

To better clarify our motivation for the two components: the input-dependent Koopman module and the U-shaped multilevel attention module, we elaborate on their design motivations in detail herein. From Figure 11, the **first** motivation is that we abandon mixture of Koopman operators and instead design an input-dependent Koopman module (KDM). Although this replacement can reduce the usage of operators and save memory occupations. However, one serious drawback is that the "input-dependent" updates tend to change smoothly and asymptotically, instead of following a regime-switching mechanism. For instance, KDM cannot shift from Pattern 1 to Pattern 2 immediately. Thus, the crucial features are missing if we only take KDM without the U-shaped multilevel attention (UMA). Second, Koopman theory was born to update the state with nonlinear dynamics by linear operations. A key assumption is that the Koopman space is infinite-dimensional. However, in practice, we set a finite Koopman space so that the approximation becomes smoother than the ground truth. Therefore, the **second** motivation to design UMA is that fine features can further be distilled from the local details. To accommodate diverse patterns, a multilevel structure provides an effective means of enhancing model flexibility.

## D. Algorithms and Complexity

We plan to publish a guiding file, all relevant codes, hyperparameter configurations on GitHub after any publication. Herein, we provide detailed pseudo-codes to support understanding of our work.

### D.1. Algorithm

Input-dependent Koopman module (KDM) is presented in Algorithm 1. Element-wise filtering is presented in Algortihm 2. Linear attention algorithm is presented in Algorithm 3.

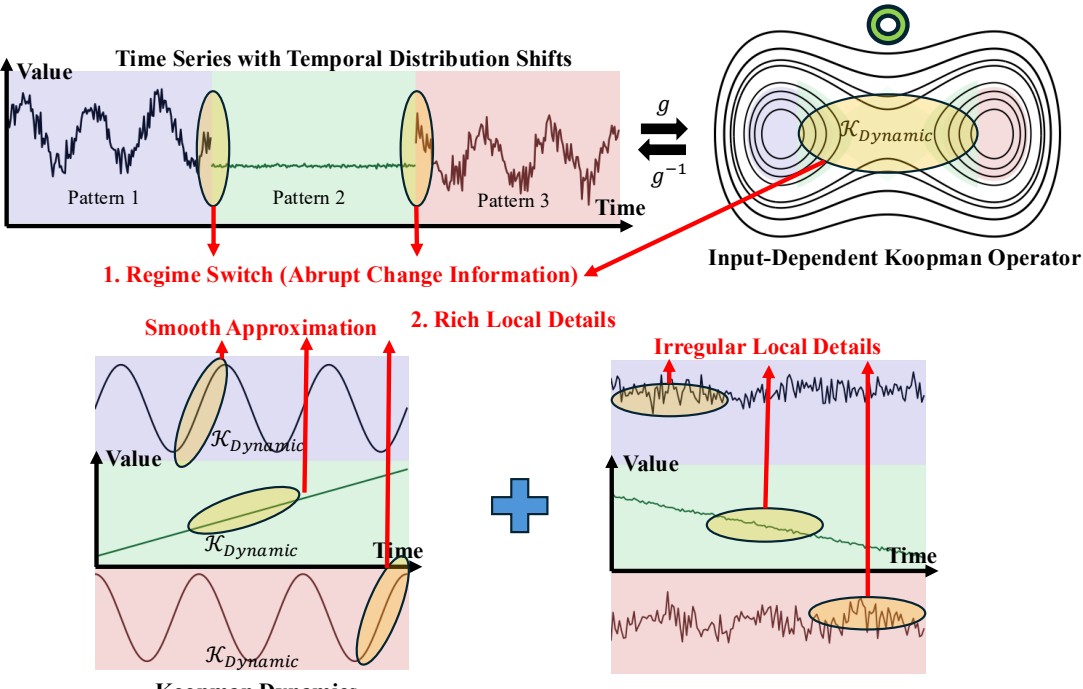

*Figure 11.* More notes marked in light yellow circles and explained by red words.

---

**Algorithm 1** Input-dependent Koopman Module

---

**Input:** $X_{emb} \in \mathbb{R}^{B \times N \times D}$, Decay ratio $r$, $R = \lfloor D/r \rfloor \in \mathbb{R}^{R \times R}$
$U \leftarrow \text{reshape}(X_{emb}W_u) \in \mathbb{R}^{B \times N \times D \times R}$; $V \leftarrow \text{reshape}(X_{emb}W_v) \in \mathbb{R}^{B \times N \times D \times R}$
$\alpha \leftarrow \text{SoftMax}\left(\frac{1}{\sqrt{D}} \sum_{d=1}^{D} V_{:,:,d,:} \odot X_{emb\,:,:,d}\right) \in \mathbb{R}^{B \times N \times R}$
$\alpha \leftarrow \alpha \odot \boldsymbol{\lambda}$ $\qquad\qquad (\boldsymbol{\lambda} \in \mathbb{R}^R)$
$\mathcal{K}_{Dynamic} \times U \leftarrow \sum_{k=1}^{R} \alpha_{:,:,k} \cdot U_{:,:,:,k} \in \mathbb{R}^{B \times N \times D}$
$K_D \leftarrow \text{Norm}\big(X_{emb} + \text{Dropout}(\text{GELU}(\mathcal{K}_{Dynamic} \times U))\big)$
**Output:** $K_D \in \mathbb{R}^{B \times N \times D}$

---

**Algorithm 2** Element-Wise Filtering

---

**Input:** $X_R \in \mathbb{R}^{B \times N \times D}$
$EQ, EK, EV \leftarrow X_R W_Q, X_R W_K, X_R W_V$
$Weighting \leftarrow \sigma\big((EQ \odot EK)/\sqrt{D}\big)$
$X_{EWF} \leftarrow Weighting \odot EV + EV$
$X_{EWF} \leftarrow \text{Dropout}(X_{EWF})$
**Output:** $X_{EWF}$

---

---

**Algorithm 3** Linear Attention

---

**Input:** $X_{EWF} \in \mathbb{R}^{B \times N \times D}$, heads $h = 4$, ROPE base $b$
$d \leftarrow \lfloor D/h \rfloor \cdot h, \quad d_h \leftarrow d/h$
$T \leftarrow \text{Linear}_{qk}(X_{EWF}) \in \mathbb{R}^{B \times N \times 2D}$
$[Q, K] \leftarrow \text{split}(T_{:,:,1:2d}); \quad V \leftarrow X_{EWF\,:,:,1:d}$
$Q \leftarrow \text{ELU}(Q) + 1; \quad K \leftarrow \text{ELU}(K) + 1$
**ROPE**$(\cdot)$ on first $d$ dims:
$\qquad \omega_j \leftarrow b^{-j/(d/2)}, \theta_{i,j} \leftarrow i\,\omega_j$; rotate each pair $(x_{2j-1}, x_{2j})$ by $(\cos(\theta_{i,j}), \sin(\theta_{i,j}))$
$Q_r \leftarrow \text{ROPE}(Q); \quad K_r \leftarrow \text{ROPE}(K)$
Reshape: $Q, Q_r, K, K_r, V \rightarrow \mathbb{R}^{B \times h \times N \times d_h}$
$Z \leftarrow \left(Q \cdot \text{mean}_n(K)\right)^{-1}$
$KV \leftarrow (K_r^\top / \sqrt{N})\,(V / \sqrt{N})$
$Y_d \leftarrow (Q_r\,KV) \odot Z; \quad Y_d \rightarrow \mathbb{R}^{B \times N \times d}$
**if** $D > d$ **then**
$\quad X_{LA} \leftarrow \text{concat}(Y_d,\ X_{EWF\,:,:,d+1:D})$
**else**
$\quad X_{LA} \leftarrow Y_d$
**end if**
**Output:** $X_{LA} + \text{Conv1D}(\text{Permute}(X_{EWF}))$

---

### D.2. Computational Complexity Process

Initially, we elaborate the alphabet signs that are adopted to compute computational complexity. $B$: Batch Size; $N$: Number of Variates; $D$ Dimension of Temporal Embeddings; $r$: Decay Rate of Koopman Space; $R$: Dimension of Koopman Space; $h$: Number of Heads; $d$: Dimension of Spited $D$ by Heads.

**Input-dependent Koopman Module (KDM)** The computational complexity of the normal multi-head attention mechanism is: $\mathcal{O}(BND^2 + BN^2D)$. This makes Transformer-based models have quadratic complexity in both temporal and variate dimensions, resulting in the issue: *High Computational Complexity*. According to Algorithm 1, the total time complexity is $\mathcal{O}(BND^3/r) = \mathcal{O}(BND^2R)$ and the total space complexity is $\mathcal{O}(BND^2/r) = \mathcal{O}(BNDR)$. The leading time complexity comes from the input-dependent operator (generation of $U, V$), which is at $\mathcal{O}(BND^2R)$. The other operations' consumptions are listed as Softmax: $\mathcal{O}(BNDR)$; $\alpha * \lambda$: $\mathcal{O}(BNR)$; $K$: $\mathcal{O}(BNDR)$; GELU/dropout/residual/RMSNorm: $\mathcal{O}(BND)$. In terms of variates $N$, KDM is linear. Compared with multi-head attention, KMD eliminates the quadratic dependency on variate amounts. While it introduces higher temporal-wise computational cost due to the expanded linear projections. The efficiency compared to Transformer-based models should be measured by the relationship between $N$ and $D^2/r$. In HVD datasets, if $N$ goes beyond $D$, KDM is more efficient than multi-head attentions. This complies with the observations in PEMS that KUMA outperforms benchmarks. In fact, the efficiency of KDM is determined by choosing an appropriate $R$. If $R$ is not dependent on $D$, KDM definitely has superiority over multi-head attentions.

**Element-Wise Filtering (EWF)** Based on Algorithm 2, the component computational complexities are all $\mathcal{O}(BND^2)$ so that the total time complexity is $\mathcal{O}(BND^2)$ and the space complexity is $\mathcal{O}(BND)$. $\mathcal{O}(BND^2)$ shakes off the quadratic curse on variates. According to fast attention mechanism, this module can be defined as linear complexity owner.

**Linear Attention (LA)** The total time complexity is $\mathcal{O}(BND^2)$ and the space complexity is $\mathcal{O}(BND + BD^2/h)$ according to Algorithm 3. The component time complexities are listed as: $Q, K$ generations: $\mathcal{O}(BND^2)$, ELU: $\mathcal{O}(BND)$, ROPE: $\mathcal{O}(BND)$, $Z$ generation: $\mathcal{O}(BNd)$, $KV$ computation: $BND^2/h$, Conv1D: $\mathcal{O}(BNkD^2)$. LA avoids $\mathcal{O}(N^2)$ and its dominate cost is $\mathcal{O}(BND^2)$, which scales linearly with $N$ for fixed $D$ and $h$.

## E. Experimental Details

### E.1. Datasets

In order to comprehensively evaluate the performance of KUMA, we conduct extensive experiments on 12 recognized time series forecasting datasets. The details of each dataset are concluded in Table 3 and the description of these datasets are

listed as below:

- Exchange-Rate (**Exchange**) collects daily exchange rates across 8 countries during 1990 to 2016 (Kitaev et al., 2020).

- **Weather** contain 21 meteorological indicators from the Weather Station of The Max Planck Biogeochemistry Institute in 2020 with a frequency at 10 minutes (Kitaev et al., 2020).

- **Solar-Energy** collects the the solar power situation of 137 PV plants in Alabama State with a frequency at 10 minutes in 2006 (Lai et al., 2018).

- Electricity (**ECL**) collects the hourly electricity consumption of 321 customers from 2012 to 2014 (Kitaev et al., 2020).

- Electricity Transformer Temperature (**ETT**) collects the load and oil temperature data of electricity transformers during July 2016 to July 2018. Four subsets are contained: ETTm1, ETTm2, ETTh1, ETTh2, where m means the frequency is at 15 minutes and h represents the frequency is at 1 hour (Kitaev et al., 2020).

- **PEMS** collects the spatial-temporal data of the public traffic networks in California. Four subsets are contained: PEMS03, PEMS04, PEMS07, PEMS08 (Chen et al., 2023).

*Table 3.* The description of all datasets employed. Variates denotes the number of variables in each dataset, Time Steps denotes the number of time points in the dataset, Granularity denotes the sampling frequency of the dataset, and Domain denotes the category information of the dataset. min represents minute(s) and h means hour(s).

| Dataset | Exchange | Weather | Solar-Energy | ECL | ETTm1 | ETTm2 | ETTh1 | ETTh2 | PEMS03 | PEMS04 | PEMS07 | PEMS08 |
|---|---|---|---|---|---|---|---|---|---|---|---|---|
| Variates | 8 | 21 | 137 | 321 | 7 | 7 | 7 | 7 | 358 | 307 | 883 | 170 |
| Time Steps | 7,588 | 52,696 | 52,560 | 26,304 | 17,420 | 17,420 | 69,680 | 69,680 | 26,209 | 16,992 | 28,224 | 17,856 |
| Granularity | 1 day | 10 min | 10 min | 1 h | 15 min | 15 min | 1 h | 1 h | 5 min | 5 min | 5 min | 5 min |
| Domain | Economy | Weather | Energy | Electricity | Electricity | Electricity | Electricity | Electricity | Traffic | Traffic | Traffic | Traffic |

### E.2. Baseline Selection Motivation

The design of KUMA is to overcome the challenges faced by Transformer-based, Linear-based, Koopman-based models, and we obtain inspiration from Mamba-based models and Multilevel-based models. Therefore, we select 15 representative models as baselines. These models have been extensively cross-tested across a wide range of prior studies and are publicly available in open-source form, ensuring both reproducibility and fair comparison.

- **Koopa** (Liu et al., 2023) adopts a mixture of Koopman operators adapting to different patterns. Since KUMA has the KDM and for fairness purpose, Koopa is selected as the candidate of Koopman-based models.

- **U-Mixer** (Ma et al., 2024) adopts both a multilevel structure and a Linear-based processor. As KUMA is of multilevel structures as well and employs linear attentions as processors. U-Mixer is selected as the representative of U-Net-inspired models.

- **TimeMixer** (Wang et al., 2024) introduces a decomposable multiscale mixing architecture to model time series patterns at different temporal resolutions. It decomposes multiscale observations into seasonal and trend components and mixes them across fine-to-coarse and coarse-to-fine directions. Therefore, we select TimeMixer as a representative multiscale baseline to evaluate whether KUMA can better balance temporal pattern extraction and forecasting efficiency.

- **SparseTSF** (Lin et al., 2026) is an extremely lightweight model that exploits cross-period sparse forecasting to capture periodic patterns with very few parameters. Due to its sparse downs-ampling strategy and its ability to decouple periodicity and trend for efficient long-term forecasting, we select SparseTSF as the candidate of sparse lightweight models and a benchmark to confirm token issue.

- **WaveRoRA** (Liang et al., 2026) is a wavelet-enhanced Transformer model that captures multivariate temporal dependencies in the wavelet domain. It introduces Rotary Route Attention (RoRA), where a small number of routing tokens aggregate information from the key-value matrices and redistribute it to the query matrix, reducing the quadratic cost of vanilla attention. Since WaveRoRA combines time-frequency representation with efficient attention to model both trends and periodic characteristics, we select it as a representative frequency-enhanced attention-based baseline. It also tests the applicability of temporal domains and frequency domains.

- **S-Mamba** (Wang et al., 2025b) is a lightweight control system that is expert at capturing long-term dependencies. Due to the fact that the input-dependent design and the linear attention are inspired by Mamba, we select S-Mamba as the candidate of Mamba-based models.

- **RLinear, DLinear, TiDE** (Zeng et al., 2023; Das et al., 2023) all employ linear projections to process either TD or VC and suffer from token scarcity. To evaluate KUMA's effectiveness on token issues, these three models are selected as benchmarks.

- **TimesNet** (Wu et al., 2023) focuses modeling TD via convolutions and is thus selected as the only TCN-based baseline.

- **iTransformer** (Liu et al., 2024) reverses the Transformer structure by encoding each individual series as variate tokens. iTransformer is one of the most advanced Transformer-based models and is selected as the baseline to confirm whether KUMA solves the challenges existing in Transformer-based models.

- **PatchTST** (Nie et al., 2023) splits the time series into patches and use original attention module to capture local details. As UMA of KUMA is designed to capture local features, PatchTST offer a fair comparison on this point.

- **Crossformer** (Zhang & Yan, 2023) segments time series data along each variate and embeds them into feature vectors. Moreover, it employs a two-stage attention mechanism to efficiently capture both TD and VC. Crossformer can act an excellent baseline to validate KUMA's ability to capture TD and VC.

- **FEDformer** (Zhou et al., 2022) captures principal components from frequency-domain and reduces computational complexity. KUMA leverages KDM to capture the principal components in time-domain. It is wise to select FEDformer as a baseline.

- **Autoformer** (Wu et al., 2021) decomposes time series into different subordinate components and learns each component individually. Similarly, KUMA separates time series into Koopman dynamics and Residual dynamics, with two carefully designed modules to process each kind of dynamics independently. Decomposition effect can be verified by comparison between Autoformer and KUMA.

### E.3. Evaluation Metrics

Following traditions in TSF, we choose Mean Squared Error (MSE) and Mean Absolute Error (MAE) as the evaluation metrics. MSE and MAE assess predictive accuracy from different perspectives. MSE emphasizes the squared errors between predictions and the ground truths, which is quite sensitive to outliers and prefers to evaluate models' ability to capture trends. By contrast, MAE measures the errors at a more macro level, which evaluates the overall performance of models. Moreover, we employ Silhouette Score measures whether samples are closer to their assigned clusters than to other clusters, thereby reflecting the clustering quality of learned representations. A higher Silhouette Score indicates stronger intra-cluster compactness and clearer inter-cluster separability. MSE, MAE, and Silhouette definitions are presented as below.

**MSE**:

$$\text{MSE} = \frac{1}{L} \sum_{i=1}^{L} (x_i - \tilde{x}_i)^2. \tag{8}$$

**MAE**:

$$\text{MAE} = \frac{1}{L} \sum_{i=1}^{L} |x_i - \tilde{x}_i|. \tag{9}$$

**Silhouette**:

$$\text{Silhouette} = \frac{1}{N} \sum_{i=1}^{N} s(i). \tag{10}$$

### E.4. Main Experiments

Due to the large number of datasets and baselines, we split the complete results into two tables. Table 4 and Table 5 present all the MSE and MAE results separately. From both tables, KUMA possesses the most best results, especially in HVD datasets. This demonstrates that KUMA can extract trends well and reduce the overall predictive errors. Besides,

Transformer-based models such as iTransformer perform well in HVD like Solar-Energy and ECL, while Linear-based models perform better in LVD like ETTh1 and ETTh2. Such observations support our claim in Section 1 and confirm the existence of token redundancy and scarcity.

*Table 4.* The complete MSE comparisons of all main experiments. All results are obtained with historical observation length $L = 12$ for PEMS datasets and $L = 96$ for the other datasets. The prediction windows are set within $\{12, 24, 48, 96\}$ for PEMS datasets and $\{96, 192, 336, 720\}$ for the rest.

| Dataset | Horizon | KUMA | Koopa | U-Mixer | WaveRoRA | TimeMixer | SparseTSF | S-Mamba | iTransformer | RLinear | PatchTST | Crossformer | TiDE | TimesNet | DLinear | FEDformer | Autoformer |
|---|---|---|---|---|---|---|---|---|---|---|---|---|---|---|---|---|---|
| | 96 | **0.085** | **0.085** | 0.087 | 0.090 | 0.096 | 0.095 | **0.086** | **0.086** | 0.093 | 0.088 | 0.256 | 0.094 | 0.107 | 0.088 | 0.148 | 0.197 |
| | 192 | 0.177 | 0.184 | 0.178 | 0.187 | **0.167** | 0.180 | 0.182 | 0.177 | 0.184 | **0.176** | 0.470 | 0.184 | 0.226 | **0.176** | 0.271 | 0.300 |
| Exchange | 336 | 0.315 | 0.327 | **0.287** | 0.347 | 0.342 | 0.321 | 0.332 | 0.331 | 0.351 | **0.301** | 1.268 | 0.349 | 0.367 | 0.313 | 0.460 | 0.509 |
| | 720 | 0.842 | 0.855 | 0.845 | 0.851 | **0.721** | **0.820** | 0.867 | 0.847 | 0.886 | 0.901 | 1.767 | 0.852 | 0.964 | 0.839 | 1.195 | 1.447 |
| | Avg | 0.355 | 0.363 | **0.349** | 0.369 | **0.332** | 0.354 | 0.367 | 0.360 | 0.378 | 0.367 | 0.940 | 0.370 | 0.416 | 0.354 | 0.519 | 0.613 |
| | 96 | **0.157** | **0.157** | 0.160 | 0.160 | 0.163 | 0.211 | 0.165 | 0.174 | 0.192 | 0.177 | **0.158** | 0.202 | 0.172 | 0.196 | 0.217 | 0.266 |
| | 192 | 0.207 | **0.196** | 0.209 | 0.210 | **0.201** | 0.263 | 0.214 | 0.221 | 0.240 | 0.225 | 0.206 | 0.242 | 0.219 | 0.237 | 0.276 | 0.307 |
| Weather | 336 | 0.268 | 0.283 | 0.268 | **0.267** | 0.258 | 0.312 | 0.274 | 0.278 | 0.292 | 0.278 | 0.272 | 0.287 | 0.280 | 0.283 | 0.339 | 0.359 |
| | 720 | 0.349 | 0.355 | 0.355 | 0.352 | **0.329** | 0.380 | 0.350 | 0.358 | 0.364 | 0.354 | 0.398 | 0.351 | 0.365 | **0.345** | 0.403 | 0.419 |
| | Avg | **0.245** | 0.248 | 0.248 | 0.247 | **0.238** | 0.291 | 0.251 | 0.258 | 0.272 | 0.259 | 0.259 | 0.271 | 0.259 | 0.265 | 0.309 | 0.338 |
| | 96 | **0.199** | 0.224 | 0.217 | **0.199** | 0.232 | 0.256 | 0.205 | **0.203** | 0.322 | 0.234 | 0.310 | 0.312 | 0.250 | 0.290 | 0.242 | 0.884 |
| | 192 | **0.232** | 0.240 | 0.237 | **0.232** | 0.238 | 0.278 | 0.237 | **0.233** | 0.359 | 0.267 | 0.734 | 0.339 | 0.296 | 0.320 | 0.285 | 0.834 |
| Solar-Energy | 336 | 0.253 | 0.255 | 0.255 | 0.251 | **0.234** | 0.294 | 0.258 | **0.248** | 0.397 | 0.290 | 0.750 | 0.368 | 0.319 | 0.353 | 0.282 | 0.941 |
| | 720 | **0.255** | 0.263 | 0.267 | 0.257 | 0.273 | 0.301 | 0.260 | **0.249** | 0.397 | 0.289 | 0.769 | 0.370 | 0.338 | 0.356 | 0.357 | 0.882 |
| | Avg | **0.235** | 0.245 | 0.244 | **0.235** | 0.244 | 0.282 | 0.240 | **0.233** | 0.369 | 0.270 | 0.641 | 0.347 | 0.301 | 0.330 | 0.291 | 0.885 |
| | 96 | **0.139** | **0.140** | 0.156 | **0.139** | 0.142 | 0.203 | **0.139** | 0.147 | 0.201 | 0.181 | 0.219 | 0.237 | 0.168 | 0.197 | 0.193 | 0.201 |
| | 192 | 0.162 | **0.158** | 0.166 | 0.160 | 0.164 | 0.209 | **0.159** | 0.162 | 0.201 | 0.188 | 0.231 | 0.236 | 0.184 | 0.196 | 0.201 | 0.222 |
| ECL | 336 | **0.174** | 0.181 | 0.182 | 0.176 | **0.171** | 0.216 | 0.176 | 0.178 | 0.215 | 0.204 | 0.246 | 0.249 | 0.198 | 0.209 | 0.214 | 0.231 |
| | 720 | **0.195** | 0.207 | 0.210 | **0.198** | 0.209 | 0.263 | 0.204 | 0.225 | 0.257 | 0.246 | 0.280 | 0.284 | 0.220 | 0.245 | 0.246 | 0.254 |
| | Avg | **0.167** | 0.172 | 0.178 | **0.168** | 0.172 | 0.223 | 0.170 | 0.178 | 0.219 | 0.205 | 0.244 | 0.251 | 0.192 | 0.212 | 0.214 | 0.227 |
| | 96 | 0.324 | **0.294** | **0.320** | 0.325 | 0.332 | 0.366 | 0.333 | 0.334 | 0.355 | 0.329 | 0.404 | 0.364 | 0.338 | 0.345 | 0.379 | 0.505 |
| | 192 | 0.364 | **0.337** | 0.369 | 0.366 | **0.355** | 0.405 | 0.376 | 0.377 | 0.391 | 0.367 | 0.450 | 0.398 | 0.374 | 0.380 | 0.426 | 0.553 |
| ETTm1 | 336 | **0.399** | 0.401 | 0.403 | 0.440 | **0.386** | 0.436 | 0.408 | 0.426 | 0.424 | **0.399** | 0.532 | 0.428 | 0.410 | 0.413 | 0.445 | 0.621 |
| | 720 | 0.463 | 0.472 | 0.461 | 0.463 | **0.452** | **0.454** | 0.475 | 0.491 | 0.487 | **0.454** | 0.666 | 0.487 | 0.478 | 0.474 | 0.543 | 0.671 |
| | Avg | 0.387 | **0.376** | 0.388 | 0.399 | **0.381** | 0.415 | 0.398 | 0.407 | 0.414 | 0.387 | 0.513 | 0.419 | 0.400 | 0.403 | 0.448 | 0.588 |
| | 96 | **0.174** | **0.175** | 0.183 | **0.175** | 0.192 | 0.201 | 0.179 | 0.180 | 0.182 | **0.175** | 0.287 | 0.207 | 0.187 | 0.193 | 0.203 | 0.255 |
| | 192 | **0.240** | 0.246 | 0.247 | 0.242 | 0.253 | 0.263 | 0.250 | 0.250 | 0.246 | **0.241** | 0.414 | 0.290 | 0.249 | 0.284 | 0.269 | 0.281 |
| ETTm2 | 336 | **0.298** | **0.305** | 0.336 | **0.305** | 0.307 | 0.316 | 0.312 | 0.311 | 0.307 | **0.305** | 0.597 | 0.377 | 0.321 | 0.369 | 0.325 | 0.399 |
| | 720 | **0.397** | 0.406 | 0.439 | 0.408 | **0.380** | 0.420 | 0.411 | 0.412 | 0.407 | 0.402 | 1.730 | 0.558 | 0.408 | 0.554 | 0.421 | 0.433 |
| | Avg | **0.277** | 0.283 | 0.301 | 0.282 | 0.283 | 0.300 | 0.288 | 0.288 | 0.286 | **0.281** | 0.757 | 0.358 | 0.291 | 0.350 | 0.305 | 0.327 |
| | 96 | **0.376** | **0.376** | 0.383 | **0.382** | 0.410 | 0.390 | 0.386 | 0.386 | 0.386 | 0.414 | 0.423 | 0.479 | 0.384 | 0.386 | **0.376** | 0.449 |
| | 192 | 0.426 | **0.422** | 0.429 | 0.429 | 0.448 | 0.440 | 0.443 | 0.441 | 0.437 | 0.460 | 0.471 | 0.525 | 0.436 | 0.437 | **0.420** | 0.500 |
| ETTh1 | 336 | 0.472 | 0.489 | 0.476 | 0.472 | 0.482 | **0.470** | 0.489 | 0.487 | 0.479 | 0.501 | 0.570 | 0.565 | 0.491 | 0.481 | **0.459** | 0.521 |
| | 720 | 0.478 | 0.492 | 0.502 | **0.470** | **0.475** | 0.476 | 0.502 | 0.503 | 0.481 | 0.500 | 0.653 | 0.594 | 0.521 | 0.519 | 0.506 | 0.514 |
| | Avg | **0.438** | 0.445 | 0.448 | **0.438** | 0.454 | 0.444 | 0.455 | 0.454 | 0.446 | 0.469 | 0.529 | 0.541 | 0.458 | 0.456 | **0.440** | 0.496 |
| | 96 | 0.290 | 0.297 | 0.290 | **0.285** | 0.315 | 0.306 | 0.296 | 0.297 | **0.288** | 0.302 | 0.745 | 0.400 | 0.340 | 0.333 | 0.358 | 0.346 |
| | 192 | **0.368** | 0.369 | 0.369 | **0.360** | 0.383 | 0.406 | 0.376 | 0.380 | 0.374 | 0.388 | 0.877 | 0.528 | 0.402 | 0.477 | 0.429 | 0.456 |
| ETTh2 | 336 | 0.415 | 0.425 | 0.425 | **0.400** | 0.385 | 0.427 | 0.424 | 0.428 | 0.415 | 0.426 | 1.043 | 0.643 | 0.452 | 0.594 | 0.496 | 0.482 |
| | 720 | 0.422 | 0.430 | 0.447 | **0.417** | 0.432 | 0.426 | 0.426 | 0.427 | **0.420** | 0.431 | 1.104 | 0.874 | 0.462 | 0.831 | 0.463 | 0.515 |
| | Avg | **0.374** | 0.380 | 0.383 | **0.365** | 0.379 | 0.391 | 0.381 | 0.383 | **0.374** | 0.387 | 0.942 | 0.611 | 0.414 | 0.559 | 0.437 | 0.450 |
| | 12 | **0.063** | 0.089 | 0.129 | 0.126 | 0.094 | 0.119 | **0.065** | 0.071 | 0.126 | 0.099 | 0.090 | 0.178 | 0.085 | 0.122 | 0.126 | 0.272 |
| | 24 | **0.082** | 0.112 | 0.256 | 0.242 | 0.135 | 0.135 | **0.087** | 0.093 | 0.246 | 0.142 | 0.121 | 0.257 | 0.118 | 0.201 | 0.149 | 0.334 |
| PEMS03 | 48 | **0.119** | 0.156 | 0.478 | 0.469 | 0.227 | 0.220 | 0.133 | **0.125** | 0.551 | 0.211 | 0.202 | 0.379 | 0.155 | 0.333 | 0.227 | 1.032 |
| | 96 | **0.173** | 0.187 | 0.462 | 0.470 | 0.289 | 0.351 | 0.201 | **0.164** | 1.057 | 0.269 | 0.262 | 0.490 | 0.228 | 0.457 | 0.348 | 1.031 |
| | Avg | **0.109** | 0.136 | 0.331 | 0.327 | 0.186 | 0.206 | 0.122 | **0.113** | 0.495 | 0.180 | 0.169 | 0.326 | 0.147 | 0.278 | 0.213 | 0.667 |
| | 12 | **0.070** | 0.083 | 0.142 | 0.146 | 0.121 | 0.130 | **0.076** | 0.078 | 0.138 | 0.105 | 0.098 | 0.219 | 0.087 | 0.148 | 0.138 | 0.424 |
| | 24 | **0.079** | 0.095 | 0.234 | 0.235 | 0.211 | 0.167 | **0.084** | 0.095 | 0.258 | 0.153 | 0.131 | 0.292 | 0.103 | 0.224 | 0.177 | 0.459 |
| PEMS04 | 48 | **0.091** | 0.128 | 0.442 | 0.461 | 0.340 | 0.254 | **0.115** | 0.120 | 0.572 | 0.229 | 0.205 | 0.409 | 0.136 | 0.355 | 0.270 | 0.646 |
| | 96 | **0.101** | 0.159 | 0.451 | 0.464 | 0.446 | 0.321 | **0.137** | 0.150 | 1.137 | 0.291 | 0.402 | 0.492 | 0.190 | 0.452 | 0.341 | 0.912 |
| | Avg | **0.085** | 0.116 | 0.317 | 0.327 | 0.280 | 0.218 | **0.103** | 0.111 | 0.526 | 0.195 | 0.209 | 0.353 | 0.129 | 0.295 | 0.231 | 0.610 |
| | 12 | **0.054** | 0.129 | 0.107 | 0.112 | 0.090 | 0.097 | **0.063** | 0.067 | 0.118 | 0.095 | 0.094 | 0.173 | 0.082 | 0.115 | 0.109 | 0.199 |
| | 24 | **0.068** | 0.236 | 0.221 | 0.227 | 0.097 | 0.115 | **0.081** | 0.088 | 0.242 | 0.150 | 0.139 | 0.271 | 0.101 | 0.210 | 0.125 | 0.323 |
| PEMS07 | 48 | **0.080** | 0.257 | 0.418 | 0.416 | 0.130 | 0.147 | **0.093** | 0.110 | 0.562 | 0.253 | 0.311 | 0.446 | 0.134 | 0.398 | 0.165 | 0.390 |
| | 96 | **0.097** | 0.259 | 0.447 | 0.450 | 0.176 | 0.255 | **0.117** | 0.139 | 1.096 | 0.346 | 0.396 | 0.628 | 0.181 | 0.594 | 0.262 | 0.554 |
| | Avg | **0.075** | 0.220 | 0.298 | 0.301 | 0.123 | 0.153 | **0.089** | 0.101 | 0.504 | 0.211 | 0.235 | 0.380 | 0.124 | 0.329 | 0.165 | 0.367 |
| | 12 | **0.070** | 0.092 | 0.137 | 0.139 | 0.141 | 0.162 | **0.076** | 0.079 | 0.133 | 0.168 | 0.165 | 0.227 | 0.112 | 0.154 | 0.173 | 0.436 |
| | 24 | **0.093** | **0.095** | 0.246 | 0.248 | 0.258 | 0.197 | 0.104 | 0.115 | 0.249 | 0.224 | 0.215 | 0.318 | 0.141 | 0.248 | 0.210 | 0.467 |
| PEMS08 | 48 | **0.137** | 0.172 | 0.442 | 0.447 | 0.413 | 0.292 | **0.167** | 0.186 | 0.569 | 0.321 | 0.315 | 0.497 | 0.198 | 0.440 | 0.320 | 0.966 |
| | 96 | **0.217** | 0.224 | 0.513 | 0.512 | 0.509 | 0.411 | 0.245 | **0.221** | 1.166 | 0.408 | 0.377 | 0.721 | 0.320 | 0.674 | 0.442 | 1.385 |
| | Avg | **0.129** | **0.146** | 0.335 | 0.337 | 0.330 | 0.266 | 0.148 | 0.150 | 0.529 | 0.280 | 0.268 | 0.441 | 0.193 | 0.379 | 0.286 | 0.814 |

**Color Marks:** **Red** = Best, **Blue** = Second Best.

## E.5. Component Effectiveness Studies

To compare the effectiveness of each component of KUMA, we first present the data employed to draw Figure 5 as Table 6. The complete results are presented in Table 7.

*Table 5.* The complete MAE comparisons of all main experiments. All results are obtained with historical observation length $L = 12$ for PEMS datasets and $L = 96$ for the other datasets. The prediction windows are set within $\{12, 24, 48, 96\}$ for PEMS datasets and $\{96, 192, 336, 720\}$ for the rest.

| Dataset | Horizon | KUMA | Koopa | U-Mixer | WaveRoRA | TimeMixer | SparseTSF | S-Mamba | iTransformer | RLinear | PatchTST | Crossformer | TiDE | TimesNet | DLinear | FEDformer | Autoformer |
|---|---|---|---|---|---|---|---|---|---|---|---|---|---|---|---|---|---|
| Exchange | 96 | 0.204 | 0.207 | 0.206 | 0.210 | 0.229 | 0.220 | 0.207 | 0.206 | 0.217 | 0.205 | 0.367 | 0.218 | 0.234 | 0.218 | 0.278 | 0.323 |
| | 192 | 0.300 | 0.309 | 0.303 | 0.309 | 0.305 | 0.304 | 0.304 | 0.299 | 0.307 | 0.299 | 0.509 | 0.307 | 0.344 | 0.315 | 0.315 | 0.369 |
| | 336 | 0.407 | 0.413 | 0.389 | 0.426 | 0.437 | 0.412 | 0.418 | 0.417 | 0.432 | 0.397 | 0.883 | 0.431 | 0.448 | 0.427 | 0.427 | 0.524 |
| | 720 | 0.690 | 0.699 | 0.697 | 0.703 | 0.734 | 0.687 | 0.703 | 0.691 | 0.714 | 0.714 | 1.068 | 0.698 | 0.746 | 0.695 | 0.695 | 0.941 |
| | Avg | 0.400 | 0.407 | 0.399 | 0.412 | 0.426 | 0.406 | 0.408 | 0.403 | 0.417 | 0.404 | 0.707 | 0.413 | 0.443 | 0.414 | 0.429 | 0.539 |
| Weather | 96 | 0.202 | 0.205 | 0.204 | 0.206 | 0.223 | 0.247 | 0.210 | 0.214 | 0.232 | 0.218 | 0.230 | 0.261 | 0.220 | 0.255 | 0.296 | 0.336 |
| | 192 | 0.249 | 0.241 | 0.253 | 0.254 | 0.254 | 0.290 | 0.252 | 0.254 | 0.271 | 0.259 | 0.277 | 0.298 | 0.261 | 0.296 | 0.336 | 0.367 |
| | 336 | 0.295 | 0.331 | 0.295 | 0.327 | 0.300 | 0.316 | 0.297 | 0.296 | 0.307 | 0.297 | 0.335 | 0.335 | 0.306 | 0.335 | 0.380 | 0.395 |
| | 720 | 0.347 | 0.347 | 0.353 | 0.347 | 0.348 | 0.361 | 0.345 | 0.347 | 0.353 | 0.348 | 0.418 | 0.386 | 0.359 | 0.381 | 0.428 | 0.428 |
| | Avg | 0.273 | 0.281 | 0.276 | 0.283 | 0.281 | 0.303 | 0.276 | 0.278 | 0.291 | 0.281 | 0.315 | 0.320 | 0.287 | 0.317 | 0.360 | 0.382 |
| Solar-Energy | 96 | 0.238 | 0.273 | 0.256 | 0.226 | 0.271 | 0.293 | 0.244 | 0.237 | 0.339 | 0.286 | 0.331 | 0.399 | 0.292 | 0.378 | 0.342 | 0.711 |
| | 192 | 0.265 | 0.275 | 0.272 | 0.261 | 0.293 | 0.317 | 0.270 | 0.261 | 0.356 | 0.310 | 0.725 | 0.416 | 0.318 | 0.398 | 0.380 | 0.692 |
| | 336 | 0.283 | 0.286 | 0.283 | 0.285 | 0.301 | 0.325 | 0.288 | 0.273 | 0.369 | 0.315 | 0.735 | 0.430 | 0.330 | 0.415 | 0.376 | 0.723 |
| | 720 | 0.283 | 0.291 | 0.289 | 0.280 | 0.319 | 0.329 | 0.288 | 0.275 | 0.356 | 0.317 | 0.765 | 0.425 | 0.337 | 0.413 | 0.427 | 0.717 |
| | Avg | 0.267 | 0.281 | 0.275 | 0.263 | 0.296 | 0.316 | 0.273 | 0.262 | 0.356 | 0.307 | 0.639 | 0.417 | 0.319 | 0.401 | 0.381 | 0.711 |
| ECL | 96 | 0.234 | 0.236 | 0.247 | 0.233 | 0.247 | 0.262 | 0.235 | 0.240 | 0.281 | 0.270 | 0.314 | 0.329 | 0.272 | 0.282 | 0.308 | 0.317 |
| | 192 | 0.257 | 0.259 | 0.259 | 0.253 | 0.275 | 0.278 | 0.255 | 0.253 | 0.283 | 0.274 | 0.322 | 0.330 | 0.289 | 0.285 | 0.315 | 0.334 |
| | 336 | 0.270 | 0.275 | 0.276 | 0.274 | 0.260 | 0.284 | 0.272 | 0.269 | 0.298 | 0.293 | 0.337 | 0.344 | 0.300 | 0.301 | 0.329 | 0.338 |
| | 720 | 0.289 | 0.312 | 0.296 | 0.293 | 0.313 | 0.327 | 0.298 | 0.317 | 0.331 | 0.324 | 0.363 | 0.373 | 0.320 | 0.333 | 0.355 | 0.361 |
| | Avg | 0.263 | 0.271 | 0.270 | 0.263 | 0.274 | 0.288 | 0.265 | 0.270 | 0.298 | 0.290 | 0.334 | 0.344 | 0.295 | 0.300 | 0.327 | 0.338 |
| ETTm1 | 96 | 0.360 | 0.345 | 0.354 | 0.359 | 0.384 | 0.371 | 0.368 | 0.368 | 0.376 | 0.367 | 0.426 | 0.387 | 0.375 | 0.372 | 0.419 | 0.475 |
| | 192 | 0.382 | 0.378 | 0.377 | 0.384 | 0.398 | 0.389 | 0.390 | 0.391 | 0.392 | 0.385 | 0.451 | 0.404 | 0.387 | 0.389 | 0.441 | 0.496 |
| | 336 | 0.406 | 0.406 | 0.409 | 0.406 | 0.416 | 0.405 | 0.413 | 0.420 | 0.415 | 0.410 | 0.515 | 0.425 | 0.411 | 0.413 | 0.459 | 0.537 |
| | 720 | 0.443 | 0.449 | 0.440 | 0.447 | 0.457 | 0.440 | 0.448 | 0.459 | 0.450 | 0.439 | 0.589 | 0.461 | 0.450 | 0.453 | 0.490 | 0.561 |
| | Avg | 0.398 | 0.395 | 0.395 | 0.399 | 0.414 | 0.401 | 0.405 | 0.410 | 0.407 | 0.400 | 0.496 | 0.419 | 0.406 | 0.407 | 0.452 | 0.517 |
| ETTm2 | 96 | 0.256 | 0.258 | 0.262 | 0.260 | 0.285 | 0.273 | 0.263 | 0.264 | 0.265 | 0.259 | 0.366 | 0.305 | 0.267 | 0.292 | 0.287 | 0.339 |
| | 192 | 0.300 | 0.302 | 0.306 | 0.304 | 0.329 | 0.310 | 0.309 | 0.309 | 0.304 | 0.302 | 0.492 | 0.364 | 0.309 | 0.362 | 0.328 | 0.340 |
| | 336 | 0.338 | 0.344 | 0.358 | 0.345 | 0.362 | 0.341 | 0.349 | 0.348 | 0.342 | 0.343 | 0.542 | 0.422 | 0.351 | 0.427 | 0.366 | 0.372 |
| | 720 | 0.396 | 0.406 | 0.423 | 0.405 | 0.412 | 0.400 | 0.406 | 0.407 | 0.398 | 0.400 | 1.042 | 0.524 | 0.403 | 0.522 | 0.415 | 0.432 |
| | Avg | 0.322 | 0.328 | 0.337 | 0.329 | 0.347 | 0.331 | 0.332 | 0.332 | 0.327 | 0.326 | 0.610 | 0.404 | 0.333 | 0.401 | 0.349 | 0.371 |
| ETTh1 | 96 | 0.399 | 0.405 | 0.396 | 0.403 | 0.441 | 0.392 | 0.405 | 0.405 | 0.395 | 0.419 | 0.448 | 0.464 | 0.402 | 0.400 | 0.419 | 0.459 |
| | 192 | 0.429 | 0.434 | 0.429 | 0.431 | 0.465 | 0.435 | 0.437 | 0.436 | 0.424 | 0.445 | 0.474 | 0.492 | 0.429 | 0.432 | 0.448 | 0.482 |
| | 336 | 0.451 | 0.460 | 0.454 | 0.458 | 0.490 | 0.442 | 0.468 | 0.458 | 0.446 | 0.466 | 0.546 | 0.515 | 0.469 | 0.459 | 0.465 | 0.496 |
| | 720 | 0.474 | 0.483 | 0.481 | 0.468 | 0.500 | 0.463 | 0.489 | 0.491 | 0.470 | 0.488 | 0.621 | 0.558 | 0.500 | 0.516 | 0.507 | 0.512 |
| | Avg | 0.438 | 0.446 | 0.440 | 0.440 | 0.474 | 0.433 | 0.450 | 0.447 | 0.434 | 0.454 | 0.522 | 0.507 | 0.450 | 0.452 | 0.460 | 0.487 |
| ETTh2 | 96 | 0.342 | 0.349 | 0.337 | 0.340 | 0.380 | 0.348 | 0.348 | 0.349 | 0.338 | 0.348 | 0.584 | 0.440 | 0.374 | 0.387 | 0.397 | 0.388 |
| | 192 | 0.392 | 0.393 | 0.393 | 0.391 | 0.415 | 0.407 | 0.396 | 0.400 | 0.390 | 0.400 | 0.656 | 0.509 | 0.414 | 0.476 | 0.439 | 0.452 |
| | 336 | 0.427 | 0.433 | 0.430 | 0.421 | 0.438 | 0.433 | 0.431 | 0.432 | 0.426 | 0.433 | 0.731 | 0.571 | 0.452 | 0.541 | 0.487 | 0.486 |
| | 720 | 0.441 | 0.444 | 0.446 | 0.437 | 0.471 | 0.445 | 0.444 | 0.445 | 0.440 | 0.446 | 0.763 | 0.679 | 0.468 | 0.657 | 0.474 | 0.511 |
| | Avg | 0.401 | 0.405 | 0.402 | 0.397 | 0.426 | 0.408 | 0.405 | 0.407 | 0.398 | 0.407 | 0.684 | 0.550 | 0.427 | 0.515 | 0.449 | 0.459 |
| PEMS03 | 12 | 0.165 | 0.207 | 0.240 | 0.245 | 0.229 | 0.247 | 0.169 | 0.174 | 0.236 | 0.216 | 0.203 | 0.305 | 0.192 | 0.243 | 0.251 | 0.385 |
| | 24 | 0.188 | 0.216 | 0.345 | 0.348 | 0.321 | 0.263 | 0.196 | 0.201 | 0.334 | 0.259 | 0.240 | 0.371 | 0.223 | 0.317 | 0.275 | 0.440 |
| | 48 | 0.228 | 0.265 | 0.472 | 0.474 | 0.535 | 0.340 | 0.243 | 0.236 | 0.529 | 0.319 | 0.317 | 0.463 | 0.260 | 0.425 | 0.348 | 0.782 |
| | 96 | 0.281 | 0.285 | 0.522 | 0.510 | 0.789 | 0.441 | 0.305 | 0.275 | 0.787 | 0.370 | 0.367 | 0.539 | 0.317 | 0.515 | 0.434 | 0.796 |
| | Avg | 0.215 | 0.243 | 0.395 | 0.394 | 0.469 | 0.323 | 0.228 | 0.221 | 0.472 | 0.291 | 0.281 | 0.419 | 0.248 | 0.375 | 0.327 | 0.601 |
| PEMS04 | 12 | 0.173 | 0.191 | 0.255 | 0.251 | 0.253 | 0.253 | 0.180 | 0.183 | 0.252 | 0.224 | 0.218 | 0.340 | 0.195 | 0.272 | 0.262 | 0.491 |
| | 24 | 0.185 | 0.210 | 0.341 | 0.337 | 0.325 | 0.271 | 0.193 | 0.205 | 0.348 | 0.275 | 0.256 | 0.398 | 0.215 | 0.340 | 0.293 | 0.509 |
| | 48 | 0.200 | 0.238 | 0.476 | 0.479 | 0.430 | 0.325 | 0.224 | 0.233 | 0.544 | 0.339 | 0.326 | 0.478 | 0.250 | 0.437 | 0.368 | 0.610 |
| | 96 | 0.212 | 0.273 | 0.500 | 0.513 | 0.455 | 0.398 | 0.248 | 0.262 | 0.820 | 0.389 | 0.457 | 0.532 | 0.303 | 0.504 | 0.427 | 0.748 |
| | Avg | 0.192 | 0.228 | 0.393 | 0.395 | 0.366 | 0.312 | 0.211 | 0.221 | 0.491 | 0.307 | 0.314 | 0.437 | 0.241 | 0.388 | 0.337 | 0.590 |
| PEMS07 | 12 | 0.147 | 0.244 | 0.236 | 0.233 | 0.251 | 0.212 | 0.159 | 0.165 | 0.235 | 0.207 | 0.200 | 0.304 | 0.181 | 0.242 | 0.225 | 0.336 |
| | 24 | 0.165 | 0.336 | 0.332 | 0.330 | 0.287 | 0.230 | 0.183 | 0.190 | 0.341 | 0.262 | 0.247 | 0.383 | 0.204 | 0.329 | 0.244 | 0.420 |
| | 48 | 0.177 | 0.344 | 0.412 | 0.407 | 0.442 | 0.260 | 0.192 | 0.215 | 0.541 | 0.340 | 0.369 | 0.495 | 0.238 | 0.458 | 0.288 | 0.470 |
| | 96 | 0.194 | 0.347 | 0.501 | 0.499 | 0.531 | 0.351 | 0.217 | 0.245 | 0.795 | 0.404 | 0.442 | 0.577 | 0.279 | 0.553 | 0.376 | 0.578 |
| | Avg | 0.171 | 0.318 | 0.370 | 0.367 | 0.378 | 0.263 | 0.188 | 0.204 | 0.478 | 0.303 | 0.315 | 0.440 | 0.225 | 0.395 | 0.283 | 0.451 |
| PEMS08 | 12 | 0.169 | 0.182 | 0.266 | 0.269 | 0.272 | 0.270 | 0.178 | 0.182 | 0.247 | 0.232 | 0.214 | 0.343 | 0.212 | 0.276 | 0.273 | 0.485 |
| | 24 | 0.193 | 0.197 | 0.334 | 0.347 | 0.352 | 0.289 | 0.209 | 0.219 | 0.343 | 0.281 | 0.260 | 0.409 | 0.238 | 0.353 | 0.301 | 0.502 |
| | 48 | 0.233 | 0.239 | 0.472 | 0.478 | 0.441 | 0.367 | 0.228 | 0.235 | 0.544 | 0.354 | 0.355 | 0.510 | 0.283 | 0.470 | 0.394 | 0.733 |
| | 96 | 0.263 | 0.266 | 0.541 | 0.551 | 0.536 | 0.423 | 0.280 | 0.267 | 0.814 | 0.417 | 0.397 | 0.592 | 0.351 | 0.565 | 0.465 | 0.915 |
| | Avg | 0.215 | 0.221 | 0.403 | 0.411 | 0.400 | 0.337 | 0.224 | 0.226 | 0.487 | 0.321 | 0.307 | 0.464 | 0.271 | 0.416 | 0.358 | 0.659 |

**Color Marks: Red** = Best, **Blue** = Second Best.

*Table 6.* The averaged comparisons of all component experiments, which is employed to draw Figure 5. All results are obtained with historical observation length $L = 12$ for PEMS datasets and $L = 96$ for the other datasets. The prediction windows are set within $\{12, 24, 48, 96\}$ for PEMS datasets and $\{96, 192, 336, 720\}$ for the rest.

| Dataset | KUMA | | UMA | | NA | | NoEWF | | NoROPE | | Mamba | | NK | |
|---|---|---|---|---|---|---|---|---|---|---|---|---|---|---|
| | MSE | MAE | MSE | MAE | MSE | MAE | MSE | MAE | MSE | MAE | MSE | MAE | MSE | MAE |
| Exchange | 0.355 | **0.400** | 0.370 | 0.410 | **0.354** | 0.404 | 0.368 | 0.415 | 0.358 | 0.404 | **0.353** | **0.399** | 0.367 | 0.410 |
| Weather | **0.245** | **0.273** | **0.242** | **0.270** | 0.257 | 0.278 | 0.254 | 0.282 | 0.246 | 0.275 | 0.257 | 0.279 | 0.248 | 0.275 |
| Solar-Energy | **0.235** | **0.267** | 0.246 | 0.280 | 0.239 | **0.269** | 0.242 | 0.278 | **0.237** | **0.269** | 0.238 | **0.267** | 0.246 | 0.281 |
| ECL | **0.167** | **0.263** | 0.173 | 0.267 | 0.185 | 0.274 | 0.174 | 0.272 | **0.170** | 0.267 | 0.171 | **0.264** | **0.170** | 0.265 |
| ETTm1 | **0.387** | **0.398** | **0.388** | **0.399** | 0.403 | 0.411 | 0.398 | 0.409 | 0.391 | 0.402 | 0.389 | 0.400 | 0.390 | 0.400 |
| ETTm2 | **0.277** | **0.322** | 0.282 | 0.326 | 0.288 | 0.333 | 0.286 | 0.332 | **0.279** | **0.325** | 0.282 | 0.327 | 0.280 | **0.325** |
| ETTh1 | **0.438** | **0.438** | 0.445 | **0.439** | 0.454 | 0.446 | 0.455 | 0.453 | **0.442** | 0.443 | **0.442** | 0.442 | 0.447 | **0.439** |
| ETTh2 | **0.374** | **0.401** | **0.374** | **0.403** | 0.393 | 0.409 | 0.388 | 0.414 | **0.378** | 0.405 | 0.380 | **0.403** | 0.389 | 0.410 |
| PEMS03 | **0.109** | **0.215** | 0.126 | 0.229 | 0.204 | 0.296 | 0.112 | 0.221 | **0.111** | **0.219** | 0.139 | 0.245 | 0.136 | 0.238 |
| PEMS04 | **0.085** | **0.192** | 0.089 | 0.199 | 0.118 | 0.228 | 0.088 | 0.197 | **0.087** | **0.196** | 0.106 | 0.216 | 0.096 | 0.207 |
| PEMS07 | **0.075** | **0.171** | 0.082 | 0.182 | 0.100 | 0.202 | 0.077 | 0.175 | **0.076** | **0.172** | 0.092 | 0.192 | 0.086 | 0.186 |
| PEMS08 | **0.129** | **0.215** | 0.154 | 0.243 | 0.308 | 0.336 | 0.134 | 0.222 | **0.131** | **0.217** | 0.157 | 0.242 | 0.164 | 0.252 |

**Color Marks: Red = Best, Blue = Second Best.**

### E.6. Robustness and Layer Influence

All the results of comparisons in Table 2 and Figure 6 are presented in Table 8. There is an obvious finding that the increase in the number of layers can contribute to better performance in HVD datasets such as PEMS03 to PEMS08. However, the enhancements are not significant, while the computational complexity rises up. The costs cannot cover the benefits. As a result, throughout the project, we employ KUMA with merely 2 layers.

## F. Additional Experiments: Longer Term Effectiveness

In addition to the ablation studies, we conduct additional experiments to explore better configuration for KUMA. We change the length of historical observations within [192, 336, 720] to discuss whether terms have impacts on KUMA. The complete results are presented in Table 9 and a more straightforward visualization is displayed in Figure 12.

From Figure 12, appropriate increases in the input length can provide more information to KUMA. This is reflected in the fact that with the input length increasing from 96 to 336, MSEs continue to reduce. The best length of historical observations is 336. However, when $L = 720$, the predictive accuracy falls down. This is caused by ancient information which having few influences on current states are also considered by KUMA. Overall, an appropriate length can boost KUMA's performance .

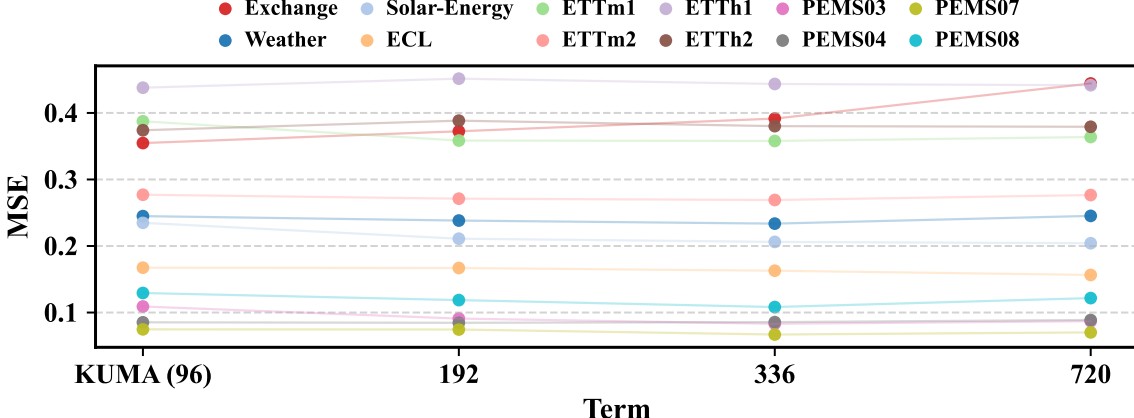

*Figure 12.* The visualization of term effects on KUMA. The lower positions of points indicate a reduction in MSE.

*Table 7.* The complete comparisons of all component experiments. All results are obtained with historical observation length $L = 12$ for PEMS datasets and $L = 96$ for the other datasets. The prediction windows are set within $\{12, 24, 48, 96\}$ for PEMS datasets and $\{96, 192, 336, 720\}$ for the rest.

| Dataset | Horizon | KUMA MSE | KUMA MAE | UMA MSE | UMA MAE | NA MSE | NA MAE | NoEWF MSE | NoEWF MAE | NoROPE MSE | NoROPE MAE | Mamba MSE | Mamba MAE | NK MSE | NK MAE |
|---|---|---|---|---|---|---|---|---|---|---|---|---|---|---|---|
| Exchange | 96 | 0.085 | 0.204 | 0.088 | 0.208 | 0.084 | 0.203 | 0.091 | 0.215 | 0.086 | 0.206 | 0.083 | 0.202 | 0.087 | 0.208 |
| | 192 | 0.177 | 0.300 | 0.184 | 0.305 | 0.187 | 0.311 | 0.183 | 0.310 | 0.179 | 0.303 | 0.177 | 0.299 | 0.188 | 0.311 |
| | 336 | 0.315 | 0.407 | 0.351 | 0.429 | 0.329 | 0.416 | 0.326 | 0.421 | 0.318 | 0.411 | 0.316 | 0.407 | 0.344 | 0.426 |
| | 720 | 0.842 | 0.690 | 0.856 | 0.699 | 0.814 | 0.683 | 0.873 | 0.713 | 0.851 | 0.697 | 0.835 | 0.688 | 0.848 | 0.695 |
| | Avg | 0.355 | 0.400 | 0.370 | 0.410 | 0.354 | 0.404 | 0.368 | 0.415 | 0.358 | 0.404 | 0.353 | 0.399 | 0.367 | 0.410 |
| Weather | 96 | 0.157 | 0.202 | 0.154 | 0.198 | 0.174 | 0.213 | 0.161 | 0.209 | 0.155 | 0.201 | 0.170 | 0.211 | 0.164 | 0.210 |
| | 192 | 0.207 | 0.249 | 0.204 | 0.246 | 0.221 | 0.255 | 0.218 | 0.260 | 0.207 | 0.250 | 0.221 | 0.256 | 0.213 | 0.252 |
| | 336 | 0.268 | 0.295 | 0.265 | 0.291 | 0.277 | 0.296 | 0.275 | 0.303 | 0.269 | 0.297 | 0.280 | 0.300 | 0.266 | 0.292 |
| | 720 | 0.349 | 0.347 | 0.346 | 0.345 | 0.355 | 0.348 | 0.359 | 0.358 | 0.354 | 0.351 | 0.357 | 0.349 | 0.347 | 0.344 |
| | Avg | 0.245 | 0.273 | 0.242 | 0.270 | 0.257 | 0.278 | 0.254 | 0.282 | 0.246 | 0.275 | 0.257 | 0.279 | 0.248 | 0.275 |
| Solar-Energy | 96 | 0.199 | 0.238 | 0.212 | 0.252 | 0.205 | 0.242 | 0.209 | 0.250 | 0.201 | 0.240 | 0.203 | 0.238 | 0.209 | 0.249 |
| | 192 | 0.232 | 0.265 | 0.239 | 0.275 | 0.239 | 0.268 | 0.239 | 0.277 | 0.236 | 0.268 | 0.236 | 0.265 | 0.239 | 0.276 |
| | 336 | 0.253 | 0.283 | 0.263 | 0.293 | 0.256 | 0.283 | 0.259 | 0.292 | 0.254 | 0.283 | 0.256 | 0.282 | 0.264 | 0.295 |
| | 720 | 0.255 | 0.283 | 0.272 | 0.300 | 0.254 | 0.284 | 0.262 | 0.294 | 0.257 | 0.286 | 0.256 | 0.284 | 0.273 | 0.302 |
| | Avg | 0.235 | 0.267 | 0.246 | 0.280 | 0.239 | 0.269 | 0.242 | 0.278 | 0.237 | 0.269 | 0.238 | 0.267 | 0.246 | 0.281 |
| ECL | 96 | 0.139 | 0.234 | 0.143 | 0.239 | 0.148 | 0.240 | 0.143 | 0.242 | 0.139 | 0.236 | 0.142 | 0.236 | 0.140 | 0.237 |
| | 192 | 0.162 | 0.257 | 0.164 | 0.259 | 0.167 | 0.258 | 0.167 | 0.263 | 0.168 | 0.262 | 0.160 | 0.252 | 0.162 | 0.257 |
| | 336 | 0.174 | 0.270 | 0.181 | 0.275 | 0.186 | 0.277 | 0.179 | 0.279 | 0.174 | 0.273 | 0.175 | 0.269 | 0.174 | 0.269 |
| | 720 | 0.195 | 0.289 | 0.204 | 0.297 | 0.238 | 0.321 | 0.207 | 0.303 | 0.199 | 0.296 | 0.205 | 0.297 | 0.205 | 0.299 |
| | Avg | 0.167 | 0.263 | 0.173 | 0.267 | 0.185 | 0.274 | 0.174 | 0.272 | 0.170 | 0.267 | 0.171 | 0.264 | 0.170 | 0.265 |
| ETTh1 | 96 | 0.376 | 0.399 | 0.382 | 0.400 | 0.386 | 0.402 | 0.398 | 0.414 | 0.380 | 0.403 | 0.378 | 0.400 | 0.391 | 0.405 |
| | 192 | 0.426 | 0.429 | 0.433 | 0.428 | 0.434 | 0.431 | 0.442 | 0.443 | 0.430 | 0.434 | 0.433 | 0.432 | 0.439 | 0.431 |
| | 336 | 0.472 | 0.451 | 0.482 | 0.454 | 0.478 | 0.455 | 0.487 | 0.465 | 0.477 | 0.456 | 0.481 | 0.458 | 0.483 | 0.452 |
| | 720 | 0.478 | 0.474 | 0.483 | 0.476 | 0.517 | 0.495 | 0.492 | 0.489 | 0.483 | 0.479 | 0.477 | 0.476 | 0.477 | 0.468 |
| | Avg | 0.438 | 0.438 | 0.445 | 0.439 | 0.454 | 0.446 | 0.455 | 0.453 | 0.442 | 0.443 | 0.442 | 0.442 | 0.447 | 0.439 |
| ETTh2 | 96 | 0.290 | 0.342 | 0.290 | 0.343 | 0.295 | 0.344 | 0.300 | 0.354 | 0.293 | 0.346 | 0.290 | 0.342 | 0.300 | 0.349 |
| | 192 | 0.368 | 0.392 | 0.368 | 0.396 | 0.384 | 0.401 | 0.381 | 0.405 | 0.372 | 0.396 | 0.371 | 0.394 | 0.376 | 0.399 |
| | 336 | 0.415 | 0.427 | 0.416 | 0.429 | 0.451 | 0.443 | 0.426 | 0.440 | 0.420 | 0.431 | 0.416 | 0.427 | 0.426 | 0.434 |
| | 720 | 0.422 | 0.441 | 0.424 | 0.443 | 0.442 | 0.447 | 0.444 | 0.458 | 0.426 | 0.443 | 0.443 | 0.449 | 0.456 | 0.457 |
| | Avg | 0.374 | 0.401 | 0.374 | 0.403 | 0.393 | 0.409 | 0.388 | 0.414 | 0.378 | 0.405 | 0.380 | 0.403 | 0.389 | 0.410 |
| ETTm1 | 96 | 0.324 | 0.360 | 0.323 | 0.362 | 0.331 | 0.369 | 0.330 | 0.369 | 0.327 | 0.364 | 0.321 | 0.359 | 0.325 | 0.363 |
| | 192 | 0.364 | 0.382 | 0.365 | 0.384 | 0.386 | 0.400 | 0.376 | 0.394 | 0.367 | 0.385 | 0.367 | 0.385 | 0.366 | 0.384 |
| | 336 | 0.399 | 0.406 | 0.401 | 0.407 | 0.412 | 0.417 | 0.409 | 0.418 | 0.403 | 0.410 | 0.401 | 0.409 | 0.402 | 0.407 |
| | 720 | 0.463 | 0.443 | 0.464 | 0.442 | 0.485 | 0.458 | 0.477 | 0.456 | 0.468 | 0.448 | 0.468 | 0.448 | 0.468 | 0.444 |
| | Avg | 0.387 | 0.398 | 0.388 | 0.399 | 0.403 | 0.411 | 0.398 | 0.409 | 0.391 | 0.402 | 0.389 | 0.400 | 0.390 | 0.400 |
| ETTm2 | 96 | 0.174 | 0.256 | 0.174 | 0.257 | 0.180 | 0.264 | 0.179 | 0.264 | 0.173 | 0.257 | 0.180 | 0.262 | 0.173 | 0.256 |
| | 192 | 0.240 | 0.300 | 0.241 | 0.301 | 0.250 | 0.311 | 0.246 | 0.309 | 0.239 | 0.301 | 0.245 | 0.305 | 0.244 | 0.304 |
| | 336 | 0.298 | 0.338 | 0.306 | 0.344 | 0.310 | 0.349 | 0.307 | 0.348 | 0.301 | 0.341 | 0.302 | 0.341 | 0.302 | 0.340 |
| | 720 | 0.397 | 0.396 | 0.405 | 0.401 | 0.411 | 0.407 | 0.412 | 0.409 | 0.405 | 0.401 | 0.401 | 0.398 | 0.403 | 0.400 |
| | Avg | 0.277 | 0.322 | 0.282 | 0.326 | 0.288 | 0.333 | 0.286 | 0.332 | 0.279 | 0.325 | 0.282 | 0.327 | 0.280 | 0.325 |
| PEMS03 | 12 | 0.063 | 0.165 | 0.064 | 0.168 | 0.071 | 0.176 | 0.065 | 0.170 | 0.063 | 0.166 | 0.066 | 0.172 | 0.068 | 0.175 |
| | 24 | 0.082 | 0.188 | 0.087 | 0.196 | 0.104 | 0.215 | 0.085 | 0.194 | 0.085 | 0.192 | 0.096 | 0.207 | 0.091 | 0.201 |
| | 48 | 0.119 | 0.228 | 0.134 | 0.239 | 0.376 | 0.428 | 0.122 | 0.233 | 0.122 | 0.232 | 0.147 | 0.258 | 0.139 | 0.245 |
| | 96 | 0.173 | 0.281 | 0.219 | 0.314 | 0.268 | 0.364 | 0.177 | 0.287 | 0.176 | 0.285 | 0.246 | 0.343 | 0.247 | 0.331 |
| | Avg | 0.109 | 0.215 | 0.126 | 0.229 | 0.204 | 0.296 | 0.112 | 0.221 | 0.111 | 0.219 | 0.139 | 0.245 | 0.136 | 0.238 |
| PEMS04 | 12 | 0.070 | 0.173 | 0.073 | 0.179 | 0.079 | 0.186 | 0.071 | 0.177 | 0.071 | 0.176 | 0.075 | 0.181 | 0.073 | 0.179 |
| | 24 | 0.079 | 0.185 | 0.081 | 0.190 | 0.099 | 0.210 | 0.081 | 0.189 | 0.081 | 0.188 | 0.090 | 0.201 | 0.088 | 0.200 |
| | 48 | 0.091 | 0.200 | 0.095 | 0.206 | 0.128 | 0.242 | 0.093 | 0.204 | 0.093 | 0.204 | 0.115 | 0.228 | 0.101 | 0.215 |
| | 96 | 0.101 | 0.212 | 0.108 | 0.220 | 0.164 | 0.275 | 0.106 | 0.219 | 0.105 | 0.216 | 0.144 | 0.256 | 0.121 | 0.236 |
| | Avg | 0.085 | 0.192 | 0.089 | 0.199 | 0.118 | 0.228 | 0.088 | 0.197 | 0.087 | 0.196 | 0.106 | 0.216 | 0.096 | 0.207 |
| PEMS07 | 12 | 0.054 | 0.147 | 0.056 | 0.153 | 0.061 | 0.158 | 0.056 | 0.150 | 0.054 | 0.147 | 0.058 | 0.154 | 0.059 | 0.154 |
| | 24 | 0.068 | 0.165 | 0.073 | 0.174 | 0.083 | 0.186 | 0.069 | 0.168 | 0.069 | 0.166 | 0.078 | 0.179 | 0.076 | 0.179 |
| | 48 | 0.080 | 0.177 | 0.091 | 0.192 | 0.114 | 0.219 | 0.084 | 0.182 | 0.083 | 0.181 | 0.102 | 0.205 | 0.096 | 0.199 |
| | 96 | 0.097 | 0.194 | 0.110 | 0.207 | 0.141 | 0.245 | 0.101 | 0.199 | 0.097 | 0.194 | 0.129 | 0.231 | 0.115 | 0.213 |
| | Avg | 0.075 | 0.171 | 0.082 | 0.182 | 0.100 | 0.202 | 0.077 | 0.175 | 0.076 | 0.172 | 0.092 | 0.192 | 0.086 | 0.186 |
| PEMS08 | 12 | 0.070 | 0.169 | 0.075 | 0.177 | 0.079 | 0.180 | 0.072 | 0.174 | 0.072 | 0.172 | 0.076 | 0.177 | 0.077 | 0.180 |
| | 24 | 0.093 | 0.193 | 0.100 | 0.204 | 0.114 | 0.215 | 0.095 | 0.199 | 0.096 | 0.196 | 0.107 | 0.210 | 0.108 | 0.219 |
| | 48 | 0.137 | 0.233 | 0.160 | 0.257 | 0.190 | 0.277 | 0.140 | 0.240 | 0.136 | 0.233 | 0.178 | 0.263 | 0.161 | 0.261 |
| | 96 | 0.217 | 0.263 | 0.282 | 0.334 | 0.848 | 0.671 | 0.228 | 0.276 | 0.221 | 0.269 | 0.265 | 0.316 | 0.308 | 0.346 |
| | Avg | 0.129 | 0.215 | 0.154 | 0.243 | 0.308 | 0.336 | 0.134 | 0.222 | 0.131 | 0.217 | 0.157 | 0.242 | 0.164 | 0.252 |
| Total Avg | | 0.240 | 0.296 | 0.248 | 0.304 | 0.275 | 0.324 | 0.248 | 0.306 | 0.242 | 0.299 | 0.250 | 0.306 | 0.252 | 0.307 |

**Color Marks: Red** = Best, **Blue** = Second Best.

*Table 8.* The complete results of KUMA under noise and with different layers. All results are obtained with historical observation length $L = 12$ for PEMS datasets and $L = 96$ for the other datasets. The prediction windows are set within $\{12, 24, 48, 96\}$ for PEMS datasets and $\{96, 192, 336, 720\}$ for the rest.

| Dataset | Horizon | KUMA (2Layer) MSE | MAE | Noise MSE | MAE | 4Layer MSE | MAE | 3Layer MSE | MAE | 1Layer MSE | MAE |
|---|---|---|---|---|---|---|---|---|---|---|---|
| | 96 | 0.085 | 0.204 | 0.093 | 0.216 | 0.085 | 0.204 | 0.090 | 0.211 | 0.085 | 0.204 |
| | 192 | 0.177 | 0.300 | 0.188 | 0.312 | 0.178 | 0.301 | 0.179 | 0.302 | 0.175 | 0.299 |
| Exchange | 336 | 0.315 | 0.407 | 0.353 | 0.431 | 0.325 | 0.414 | 0.326 | 0.415 | 0.327 | 0.415 |
| | 720 | 0.842 | 0.690 | 0.892 | 0.713 | 0.848 | 0.693 | 0.841 | 0.690 | 0.845 | 0.691 |
| | Avg | 0.355 | 0.400 | 0.382 | 0.418 | 0.359 | 0.403 | 0.359 | 0.405 | 0.358 | 0.402 |
| | 96 | 0.157 | 0.202 | 0.156 | 0.207 | 0.158 | 0.203 | 0.159 | 0.205 | 0.163 | 0.208 |
| | 192 | 0.207 | 0.249 | 0.209 | 0.257 | 0.209 | 0.251 | 0.205 | 0.248 | 0.213 | 0.254 |
| Weather | 336 | 0.268 | 0.295 | 0.268 | 0.301 | 0.273 | 0.296 | 0.272 | 0.295 | 0.270 | 0.293 |
| | 720 | 0.349 | 0.347 | 0.354 | 0.358 | 0.350 | 0.349 | 0.345 | 0.345 | 0.349 | 0.346 |
| | Avg | 0.245 | 0.273 | 0.246 | 0.281 | 0.247 | 0.275 | 0.245 | 0.273 | 0.249 | 0.275 |
| | 96 | 0.199 | 0.238 | 0.190 | 0.246 | 0.197 | 0.239 | 0.198 | 0.238 | 0.200 | 0.240 |
| | 192 | 0.232 | 0.265 | 0.234 | 0.285 | 0.242 | 0.274 | 0.233 | 0.267 | 0.234 | 0.268 |
| Solar-Energy | 336 | 0.253 | 0.283 | 0.244 | 0.289 | 0.251 | 0.281 | 0.253 | 0.282 | 0.250 | 0.281 |
| | 720 | 0.255 | 0.283 | 0.234 | 0.284 | 0.252 | 0.284 | 0.252 | 0.284 | 0.254 | 0.284 |
| | Avg | 0.235 | 0.267 | 0.225 | 0.276 | 0.236 | 0.269 | 0.234 | 0.268 | 0.234 | 0.268 |
| | 96 | 0.139 | 0.234 | 0.140 | 0.239 | 0.137 | 0.232 | 0.139 | 0.234 | 0.138 | 0.234 |
| | 192 | 0.162 | 0.257 | 0.164 | 0.261 | 0.170 | 0.260 | 0.164 | 0.258 | 0.161 | 0.253 |
| ECL | 336 | 0.174 | 0.270 | 0.175 | 0.274 | 0.177 | 0.271 | 0.174 | 0.270 | 0.178 | 0.272 |
| | 720 | 0.195 | 0.289 | 0.193 | 0.292 | 0.213 | 0.300 | 0.202 | 0.294 | 0.204 | 0.297 |
| | Avg | 0.167 | 0.263 | 0.168 | 0.267 | 0.174 | 0.266 | 0.170 | 0.264 | 0.170 | 0.264 |
| | 96 | 0.324 | 0.360 | 0.318 | 0.357 | 0.323 | 0.359 | 0.322 | 0.358 | 0.321 | 0.358 |
| | 192 | 0.364 | 0.382 | 0.363 | 0.381 | 0.366 | 0.383 | 0.366 | 0.383 | 0.364 | 0.382 |
| ETTm1 | 336 | 0.399 | 0.406 | 0.396 | 0.403 | 0.397 | 0.405 | 0.397 | 0.405 | 0.398 | 0.405 |
| | 720 | 0.463 | 0.443 | 0.457 | 0.437 | 0.462 | 0.443 | 0.465 | 0.443 | 0.463 | 0.443 |
| | Avg | 0.387 | 0.398 | 0.384 | 0.395 | 0.387 | 0.398 | 0.387 | 0.397 | 0.386 | 0.397 |
| | 96 | 0.174 | 0.256 | 0.174 | 0.260 | 0.175 | 0.256 | 0.173 | 0.256 | 0.173 | 0.255 |
| | 192 | 0.240 | 0.300 | 0.242 | 0.304 | 0.240 | 0.300 | 0.239 | 0.299 | 0.239 | 0.299 |
| ETTm2 | 336 | 0.298 | 0.338 | 0.297 | 0.339 | 0.300 | 0.339 | 0.298 | 0.338 | 0.300 | 0.339 |
| | 720 | 0.397 | 0.396 | 0.396 | 0.397 | 0.397 | 0.396 | 0.400 | 0.398 | 0.395 | 0.394 |
| | Avg | 0.277 | 0.322 | 0.277 | 0.325 | 0.278 | 0.323 | 0.278 | 0.322 | 0.277 | 0.322 |
| | 96 | 0.376 | 0.399 | 0.377 | 0.398 | 0.384 | 0.401 | 0.380 | 0.400 | 0.390 | 0.403 |
| | 192 | 0.426 | 0.429 | 0.428 | 0.428 | 0.436 | 0.433 | 0.433 | 0.432 | 0.436 | 0.433 |
| ETTh1 | 336 | 0.472 | 0.451 | 0.467 | 0.447 | 0.474 | 0.452 | 0.465 | 0.448 | 0.472 | 0.451 |
| | 720 | 0.478 | 0.474 | 0.467 | 0.466 | 0.488 | 0.479 | 0.477 | 0.476 | 0.477 | 0.476 |
| | Avg | 0.438 | 0.438 | 0.435 | 0.435 | 0.446 | 0.441 | 0.439 | 0.439 | 0.444 | 0.441 |
| | 96 | 0.290 | 0.342 | 0.291 | 0.347 | 0.299 | 0.348 | 0.291 | 0.343 | 0.294 | 0.344 |
| | 192 | 0.368 | 0.392 | 0.367 | 0.395 | 0.378 | 0.400 | 0.373 | 0.395 | 0.369 | 0.393 |
| ETTh2 | 336 | 0.415 | 0.427 | 0.414 | 0.429 | 0.415 | 0.430 | 0.423 | 0.431 | 0.428 | 0.433 |
| | 720 | 0.422 | 0.441 | 0.419 | 0.441 | 0.432 | 0.446 | 0.427 | 0.446 | 0.421 | 0.441 |
| | Avg | 0.374 | 0.401 | 0.373 | 0.403 | 0.381 | 0.406 | 0.379 | 0.404 | 0.378 | 0.403 |
| | 12 | 0.063 | 0.165 | 0.063 | 0.166 | 0.062 | 0.165 | 0.063 | 0.165 | 0.063 | 0.166 |
| | 24 | 0.082 | 0.188 | 0.083 | 0.190 | 0.083 | 0.189 | 0.081 | 0.188 | 0.083 | 0.190 |
| PEMS03 | 48 | 0.119 | 0.228 | 0.121 | 0.229 | 0.125 | 0.231 | 0.123 | 0.231 | 0.124 | 0.232 |
| | 96 | 0.173 | 0.281 | 0.176 | 0.284 | 0.167 | 0.275 | 0.169 | 0.278 | 0.170 | 0.279 |
| | Avg | 0.109 | 0.215 | 0.111 | 0.217 | 0.109 | 0.215 | 0.109 | 0.215 | 0.110 | 0.216 |
| | 12 | 0.070 | 0.173 | 0.072 | 0.177 | 0.069 | 0.171 | 0.069 | 0.172 | 0.072 | 0.177 |
| | 24 | 0.079 | 0.185 | 0.081 | 0.188 | 0.077 | 0.182 | 0.077 | 0.183 | 0.082 | 0.189 |
| PEMS04 | 48 | 0.091 | 0.200 | 0.090 | 0.198 | 0.090 | 0.198 | 0.090 | 0.198 | 0.092 | 0.201 |
| | 96 | 0.101 | 0.212 | 0.102 | 0.211 | 0.100 | 0.210 | 0.103 | 0.212 | 0.105 | 0.215 |
| | Avg | 0.085 | 0.192 | 0.086 | 0.193 | 0.084 | 0.190 | 0.085 | 0.191 | 0.088 | 0.196 |
| | 12 | 0.054 | 0.147 | 0.054 | 0.147 | 0.054 | 0.146 | 0.055 | 0.147 | 0.055 | 0.149 |
| | 24 | 0.068 | 0.165 | 0.070 | 0.167 | 0.067 | 0.163 | 0.067 | 0.164 | 0.070 | 0.169 |
| PEMS07 | 48 | 0.080 | 0.177 | 0.082 | 0.180 | 0.079 | 0.176 | 0.080 | 0.176 | 0.082 | 0.181 |
| | 96 | 0.097 | 0.194 | 0.098 | 0.194 | 0.096 | 0.191 | 0.097 | 0.193 | 0.100 | 0.197 |
| | Avg | 0.075 | 0.171 | 0.076 | 0.172 | 0.074 | 0.169 | 0.075 | 0.170 | 0.077 | 0.174 |
| | 12 | 0.070 | 0.169 | 0.073 | 0.174 | 0.071 | 0.169 | 0.072 | 0.170 | 0.071 | 0.172 |
| | 24 | 0.093 | 0.193 | 0.096 | 0.199 | 0.090 | 0.190 | 0.092 | 0.191 | 0.095 | 0.197 |
| PEMS08 | 48 | 0.137 | 0.233 | 0.136 | 0.237 | 0.136 | 0.229 | 0.130 | 0.228 | 0.135 | 0.231 |
| | 96 | 0.217 | 0.263 | 0.226 | 0.280 | 0.228 | 0.271 | 0.202 | 0.261 | 0.196 | 0.263 |
| | Avg | 0.129 | 0.215 | 0.133 | 0.222 | 0.131 | 0.215 | 0.124 | 0.213 | 0.124 | 0.215 |
| Total Avg | | 0.240 | 0.296 | 0.241 | 0.300 | 0.242 | 0.297 | 0.240 | 0.297 | 0.241 | 0.298 |

**Color Marks: Red** = Best, **Blue** = Second Best.

*Table 9.* The complete results of KUMA with different $L$ lengths of historical observations. All results are obtained with historical observation length $L = 12$ for PEMS datasets and $L = 96$ for the other datasets. The prediction windows are set within $\{12, 24, 48, 96\}$ for PEMS datasets and $\{96, 192, 336, 720\}$ for the rest.

| Dataset | Horizon | KUMA (96) MSE | MAE | 192 MSE | MAE | 336 MSE | MAE | 720 MSE | MAE |
|---|---|---|---|---|---|---|---|---|---|
| Exchange | 96 | 0.085 | 0.204 | 0.097 | 0.220 | 0.096 | 0.220 | 0.092 | 0.217 |
| | 192 | 0.177 | 0.300 | 0.188 | 0.311 | 0.190 | 0.314 | 0.197 | 0.325 |
| | 336 | 0.315 | 0.407 | 0.329 | 0.417 | 0.366 | 0.441 | 0.392 | 0.452 |
| | 720 | 0.842 | 0.690 | 0.876 | 0.702 | 0.914 | 0.715 | 1.097 | 0.778 |
| | Avg | 0.355 | 0.400 | 0.372 | 0.412 | 0.392 | 0.423 | 0.444 | 0.443 |
| Weather | 96 | 0.157 | 0.202 | 0.159 | 0.205 | 0.154 | 0.204 | 0.164 | 0.218 |
| | 192 | 0.207 | 0.249 | 0.202 | 0.245 | 0.200 | 0.248 | 0.213 | 0.261 |
| | 336 | 0.268 | 0.295 | 0.259 | 0.289 | 0.249 | 0.286 | 0.266 | 0.301 |
| | 720 | 0.349 | 0.347 | 0.334 | 0.340 | 0.332 | 0.342 | 0.338 | 0.350 |
| | Avg | 0.245 | 0.273 | 0.238 | 0.270 | 0.234 | 0.270 | 0.245 | 0.283 |
| Solar-Energy | 96 | 0.199 | 0.238 | 0.189 | 0.236 | 0.195 | 0.243 | 0.189 | 0.250 |
| | 192 | 0.232 | 0.265 | 0.214 | 0.260 | 0.214 | 0.263 | 0.198 | 0.261 |
| | 336 | 0.253 | 0.283 | 0.217 | 0.264 | 0.207 | 0.260 | 0.217 | 0.269 |
| | 720 | 0.255 | 0.283 | 0.224 | 0.272 | 0.209 | 0.265 | 0.213 | 0.271 |
| | Avg | 0.235 | 0.267 | 0.211 | 0.258 | 0.206 | 0.258 | 0.204 | 0.263 |
| ECL | 96 | 0.139 | 0.234 | 0.136 | 0.230 | 0.135 | 0.229 | 0.132 | 0.225 |
| | 192 | 0.162 | 0.257 | 0.162 | 0.253 | 0.154 | 0.249 | 0.149 | 0.244 |
| | 336 | 0.174 | 0.270 | 0.177 | 0.269 | 0.168 | 0.262 | 0.160 | 0.259 |
| | 720 | 0.195 | 0.289 | 0.192 | 0.289 | 0.194 | 0.288 | 0.185 | 0.288 |
| | Avg | 0.167 | 0.263 | 0.167 | 0.260 | 0.163 | 0.257 | 0.156 | 0.254 |
| ETTh1 | 96 | 0.376 | 0.399 | 0.389 | 0.407 | 0.393 | 0.413 | 0.406 | 0.424 |
| | 192 | 0.426 | 0.429 | 0.439 | 0.436 | 0.431 | 0.436 | 0.427 | 0.443 |
| | 336 | 0.472 | 0.451 | 0.472 | 0.455 | 0.451 | 0.451 | 0.457 | 0.463 |
| | 720 | 0.478 | 0.474 | 0.506 | 0.495 | 0.499 | 0.495 | 0.476 | 0.485 |
| | Avg | 0.438 | 0.438 | 0.452 | 0.448 | 0.444 | 0.449 | 0.442 | 0.454 |
| ETTh2 | 96 | 0.290 | 0.342 | 0.309 | 0.358 | 0.306 | 0.360 | 0.313 | 0.361 |
| | 192 | 0.368 | 0.392 | 0.385 | 0.404 | 0.382 | 0.406 | 0.370 | 0.403 |
| | 336 | 0.415 | 0.427 | 0.410 | 0.426 | 0.404 | 0.429 | 0.391 | 0.424 |
| | 720 | 0.422 | 0.441 | 0.450 | 0.455 | 0.429 | 0.449 | 0.443 | 0.460 |
| | Avg | 0.374 | 0.401 | 0.388 | 0.411 | 0.380 | 0.411 | 0.379 | 0.412 |
| ETTm1 | 96 | 0.324 | 0.360 | 0.301 | 0.351 | 0.295 | 0.346 | 0.309 | 0.357 |
| | 192 | 0.364 | 0.382 | 0.338 | 0.372 | 0.334 | 0.374 | 0.344 | 0.381 |
| | 336 | 0.399 | 0.406 | 0.365 | 0.390 | 0.374 | 0.397 | 0.371 | 0.397 |
| | 720 | 0.463 | 0.443 | 0.430 | 0.428 | 0.428 | 0.429 | 0.430 | 0.431 |
| | Avg | 0.387 | 0.398 | 0.358 | 0.385 | 0.358 | 0.387 | 0.364 | 0.392 |
| ETTm2 | 96 | 0.174 | 0.256 | 0.174 | 0.258 | 0.179 | 0.261 | 0.181 | 0.267 |
| | 192 | 0.240 | 0.300 | 0.235 | 0.299 | 0.241 | 0.305 | 0.247 | 0.315 |
| | 336 | 0.298 | 0.338 | 0.291 | 0.338 | 0.285 | 0.337 | 0.302 | 0.349 |
| | 720 | 0.397 | 0.396 | 0.384 | 0.394 | 0.371 | 0.390 | 0.376 | 0.397 |
| | Avg | 0.277 | 0.322 | 0.271 | 0.322 | 0.269 | 0.323 | 0.277 | 0.332 |
| PEMS03 | 12 | 0.063 | 0.165 | 0.060 | 0.164 | 0.057 | 0.157 | 0.061 | 0.164 |
| | 24 | 0.082 | 0.188 | 0.077 | 0.187 | 0.072 | 0.173 | 0.076 | 0.181 |
| | 48 | 0.119 | 0.228 | 0.103 | 0.215 | 0.091 | 0.196 | 0.098 | 0.202 |
| | 96 | 0.173 | 0.281 | 0.124 | 0.235 | 0.112 | 0.214 | 0.113 | 0.216 |
| | Avg | 0.109 | 0.215 | 0.091 | 0.200 | 0.083 | 0.185 | 0.087 | 0.191 |
| PEMS04 | 12 | 0.070 | 0.173 | 0.070 | 0.173 | 0.070 | 0.172 | 0.074 | 0.178 |
| | 24 | 0.079 | 0.185 | 0.077 | 0.183 | 0.079 | 0.183 | 0.081 | 0.185 |
| | 48 | 0.091 | 0.200 | 0.090 | 0.198 | 0.091 | 0.197 | 0.091 | 0.195 |
| | 96 | 0.101 | 0.212 | 0.101 | 0.209 | 0.102 | 0.205 | 0.109 | 0.212 |
| | Avg | 0.085 | 0.192 | 0.085 | 0.191 | 0.086 | 0.189 | 0.089 | 0.193 |
| PEMS07 | 12 | 0.054 | 0.147 | 0.055 | 0.146 | 0.052 | 0.141 | 0.054 | 0.142 |
| | 24 | 0.068 | 0.165 | 0.068 | 0.162 | 0.062 | 0.155 | 0.064 | 0.155 |
| | 48 | 0.080 | 0.177 | 0.079 | 0.174 | 0.072 | 0.163 | 0.074 | 0.166 |
| | 96 | 0.097 | 0.194 | 0.096 | 0.186 | 0.082 | 0.172 | 0.088 | 0.175 |
| | Avg | 0.075 | 0.171 | 0.074 | 0.167 | 0.067 | 0.158 | 0.070 | 0.160 |
| PEMS08 | 12 | 0.070 | 0.169 | 0.074 | 0.172 | 0.066 | 0.161 | 0.072 | 0.166 |
| | 24 | 0.093 | 0.193 | 0.092 | 0.188 | 0.080 | 0.173 | 0.099 | 0.190 |
| | 48 | 0.137 | 0.233 | 0.128 | 0.216 | 0.103 | 0.188 | 0.138 | 0.222 |
| | 96 | 0.217 | 0.263 | 0.181 | 0.228 | 0.184 | 0.221 | 0.177 | 0.215 |
| | Avg | 0.129 | 0.215 | 0.119 | 0.201 | 0.108 | 0.186 | 0.122 | 0.198 |
| Total Avg | | 0.240 | 0.296 | 0.236 | 0.294 | 0.232 | 0.291 | 0.240 | 0.298 |

**Color Marks: Red** = Best, **Blue** = Second Best.

## G. Statistical Test

Three statistical tests are conducted in this work to assess the robustness and significance of the forecasting results. The t-test results are reported in Table 10, while the Friedman nonparametric test and Wilcoxon signed-rank test results are presented in Table 11 and Table 12, respectively.

## H. Separation Visualization

To demonstrate the effectiveness of KDM, we first visualize the separated representations using UMAP and quantify their separability with Silhouette scores. An example on ETTh2 is shown in Fig. 13. Based on this separation, we further present the corresponding component signals in Fig. 14.

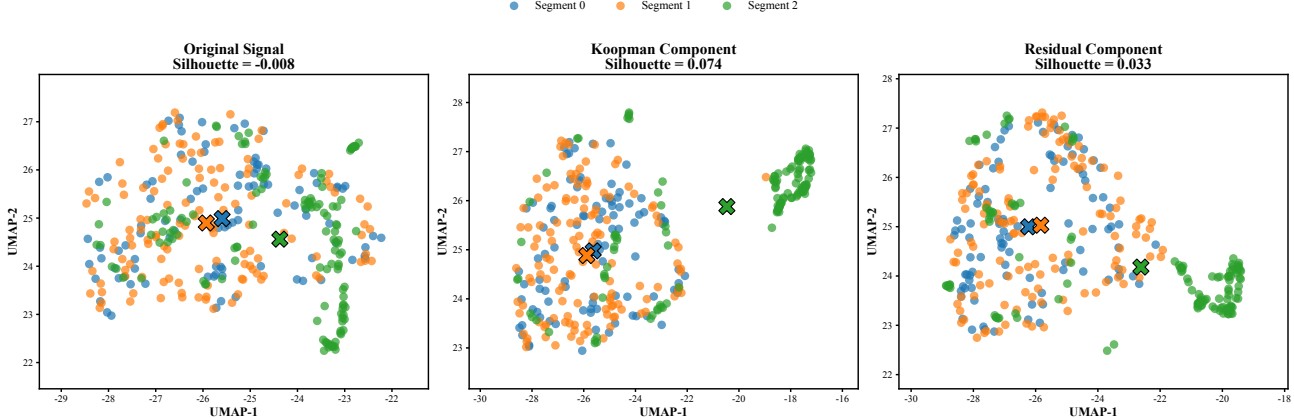

*Figure 13.* An example of UMAP of KUMA to visualize the separation effectiveness on ETTh2 with 96 historical observations for 96 forecasting windows.

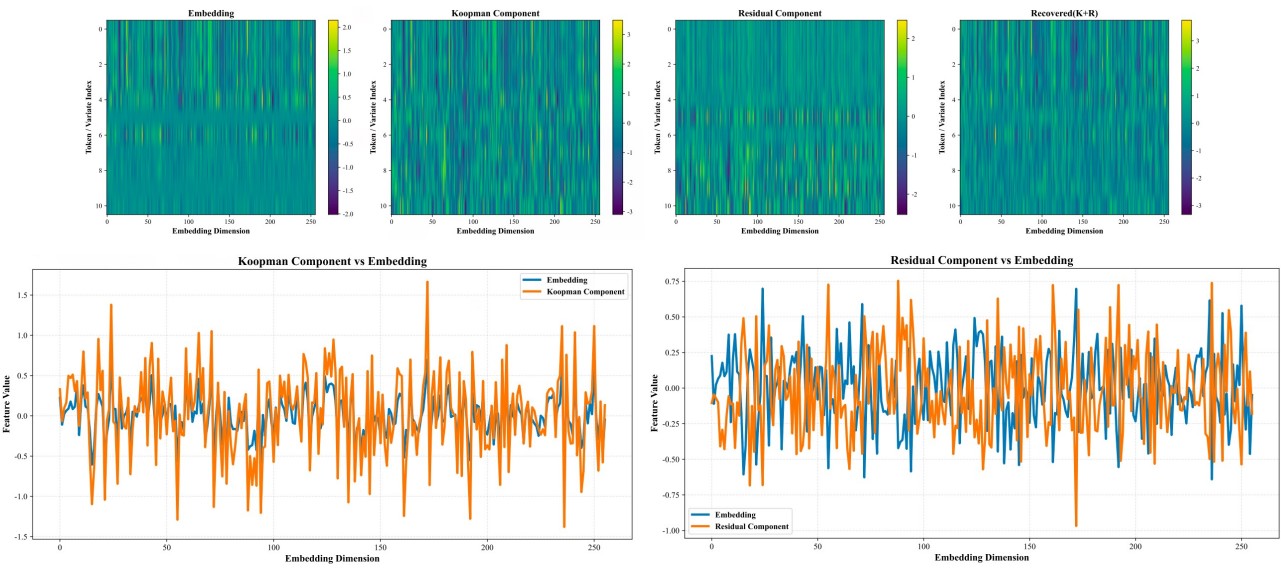

*Figure 14.* An example of KUMA on ETTh2, which shows the different components and their heatmaps.

*Table 10.* Mean, standard deviation, and 95% confidence interval of MSE and MAE across repeated runs.

| Dataset | Horizon | MSE | | | MAE | | |
|---|---|---|---|---|---|---|---|
| | | Mean | Std | 95% CI | Mean | Std | 95% CI |
| Exchange | 96 | 0.087 | 0.002 | [0.084, 0.089] | 0.206 | 0.003 | [0.203, 0.209] |
| | 192 | 0.179 | 0.004 | [0.175, 0.183] | 0.301 | 0.002 | [0.299, 0.304] |
| | 336 | 0.345 | 0.030 | [0.313, 0.377] | 0.426 | 0.019 | [0.407, 0.446] |
| | 720 | 0.846 | 0.010 | [0.836, 0.856] | 0.692 | 0.003 | [0.689, 0.695] |
| | Avg | 0.364 | 0.006 | [0.357, 0.371] | 0.407 | 0.004 | [0.402, 0.411] |
| Weather | 96 | 0.160 | 0.002 | [0.158, 0.162] | 0.205 | 0.002 | [0.203, 0.207] |
| | 192 | 0.208 | 0.002 | [0.206, 0.210] | 0.250 | 0.002 | [0.248, 0.251] |
| | 336 | 0.268 | 0.002 | [0.267, 0.270] | 0.295 | 0.002 | [0.293, 0.296] |
| | 720 | 0.349 | 0.002 | [0.347, 0.351] | 0.347 | 0.002 | [0.346, 0.349] |
| | Avg | 0.246 | 0.001 | [0.245, 0.247] | 0.274 | 0.001 | [0.273, 0.275] |
| Solar-Energy | 96 | 0.198 | 0.001 | [0.197, 0.200] | 0.239 | 0.002 | [0.237, 0.241] |
| | 192 | 0.234 | 0.002 | [0.232, 0.236] | 0.266 | 0.002 | [0.264, 0.268] |
| | 336 | 0.252 | 0.001 | [0.251, 0.254] | 0.283 | 0.001 | [0.282, 0.283] |
| | 720 | 0.253 | 0.001 | [0.252, 0.254] | 0.284 | 0.000 | [0.283, 0.284] |
| | Avg | 0.234 | 0.001 | [0.233, 0.235] | 0.268 | 0.001 | [0.267, 0.269] |
| ECL | 96 | 0.139 | 0.001 | [0.137, 0.140] | 0.234 | 0.001 | [0.232, 0.235] |
| | 192 | 0.163 | 0.001 | [0.162, 0.164] | 0.256 | 0.001 | [0.255, 0.257] |
| | 336 | 0.182 | 0.007 | [0.174, 0.189] | 0.276 | 0.005 | [0.271, 0.282] |
| | 720 | 0.203 | 0.008 | [0.195, 0.212] | 0.296 | 0.005 | [0.290, 0.301] |
| | Avg | 0.172 | 0.003 | [0.168, 0.175] | 0.265 | 0.002 | [0.263, 0.268] |
| ETTm1 | 96 | 0.325 | 0.004 | [0.320, 0.329] | 0.361 | 0.003 | [0.358, 0.364] |
| | 192 | 0.364 | 0.001 | [0.363, 0.365] | 0.382 | 0.001 | [0.381, 0.382] |
| | 336 | 0.397 | 0.001 | [0.396, 0.398] | 0.405 | 0.001 | [0.404, 0.406] |
| | 720 | 0.463 | 0.001 | [0.462, 0.463] | 0.442 | 0.000 | [0.442, 0.443] |
| | Avg | 0.387 | 0.001 | [0.386, 0.388] | 0.398 | 0.001 | [0.397, 0.398] |
| ETTm2 | 96 | 0.174 | 0.000 | [0.173, 0.174] | 0.255 | 0.000 | [0.255, 0.256] |
| | 192 | 0.239 | 0.001 | [0.238, 0.240] | 0.300 | 0.000 | [0.299, 0.300] |
| | 336 | 0.300 | 0.001 | [0.298, 0.301] | 0.338 | 0.001 | [0.338, 0.339] |
| | 720 | 0.397 | 0.001 | [0.395, 0.398] | 0.396 | 0.001 | [0.395, 0.397] |
| | Avg | 0.277 | 0.000 | [0.277, 0.278] | 0.322 | 0.000 | [0.322, 0.323] |
| ETTh1 | 96 | 0.380 | 0.004 | [0.377, 0.384] | 0.401 | 0.002 | [0.399, 0.402] |
| | 192 | 0.433 | 0.004 | [0.429, 0.437] | 0.432 | 0.001 | [0.430, 0.433] |
| | 336 | 0.471 | 0.002 | [0.469, 0.473] | 0.451 | 0.001 | [0.450, 0.453] |
| | 720 | 0.484 | 0.006 | [0.478, 0.490] | 0.478 | 0.003 | [0.474, 0.481] |
| | Avg | 0.442 | 0.002 | [0.440, 0.445] | 0.440 | 0.001 | [0.439, 0.442] |
| ETTh2 | 96 | 0.295 | 0.005 | [0.290, 0.300] | 0.345 | 0.003 | [0.342, 0.349] |
| | 192 | 0.371 | 0.003 | [0.368, 0.374] | 0.393 | 0.003 | [0.390, 0.396] |
| | 336 | 0.420 | 0.003 | [0.417, 0.423] | 0.430 | 0.002 | [0.428, 0.431] |
| | 720 | 0.430 | 0.011 | [0.418, 0.442] | 0.445 | 0.005 | [0.440, 0.450] |
| | Avg | 0.379 | 0.004 | [0.375, 0.383] | 0.403 | 0.002 | [0.401, 0.406] |
| PEMS03 | 12 | 0.063 | 0.001 | [0.062, 0.064] | 0.166 | 0.001 | [0.165, 0.167] |
| | 24 | 0.082 | 0.001 | [0.082, 0.083] | 0.189 | 0.001 | [0.188, 0.190] |
| | 48 | 0.122 | 0.003 | [0.119, 0.125] | 0.230 | 0.002 | [0.227, 0.232] |
| | 96 | 0.176 | 0.013 | [0.162, 0.190] | 0.283 | 0.009 | [0.273, 0.292] |
| | Avg | 0.111 | 0.004 | [0.107, 0.115] | 0.217 | 0.003 | [0.214, 0.220] |
| PEMS04 | 12 | 0.070 | 0.000 | [0.070, 0.071] | 0.173 | 0.001 | [0.173, 0.174] |
| | 24 | 0.078 | 0.000 | [0.078, 0.079] | 0.185 | 0.001 | [0.184, 0.185] |
| | 48 | 0.091 | 0.001 | [0.090, 0.092] | 0.200 | 0.002 | [0.198, 0.201] |
| | 96 | 0.103 | 0.001 | [0.102, 0.105] | 0.214 | 0.002 | [0.212, 0.216] |
| | Avg | 0.086 | 0.000 | [0.085, 0.086] | 0.193 | 0.001 | [0.192, 0.194] |
| PEMS07 | 12 | 0.054 | 0.001 | [0.054, 0.055] | 0.147 | 0.000 | [0.146, 0.147] |
| | 24 | 0.068 | 0.000 | [0.067, 0.068] | 0.165 | 0.000 | [0.164, 0.165] |
| | 48 | 0.081 | 0.001 | [0.080, 0.081] | 0.177 | 0.001 | [0.177, 0.178] |
| | 96 | 0.098 | 0.001 | [0.097, 0.099] | 0.194 | 0.002 | [0.192, 0.196] |
| | Avg | 0.075 | 0.000 | [0.075, 0.076] | 0.171 | 0.000 | [0.170, 0.171] |
| PEMS08 | 12 | 0.071 | 0.000 | [0.070, 0.071] | 0.170 | 0.000 | [0.170, 0.170] |
| | 24 | 0.092 | 0.001 | [0.091, 0.094] | 0.193 | 0.001 | [0.192, 0.194] |
| | 48 | 0.139 | 0.002 | [0.136, 0.142] | 0.233 | 0.001 | [0.231, 0.234] |
| | 96 | 0.213 | 0.005 | [0.208, 0.219] | 0.265 | 0.006 | [0.259, 0.271] |
| | Avg | 0.129 | 0.001 | [0.127, 0.130] | 0.215 | 0.002 | [0.213, 0.217] |
| Total Avg | | 0.242 | 0.001 | [0.241, 0.243] | 0.298 | 0.001 | [0.297, 0.299] |

Mean and standard deviation are computed across repeated runs. CI denotes the two-sided 95% confidence interval, calculated as $\bar{x} \pm t_{0.975, n-1} s/\sqrt{n}$.

*Table 11.* The Friedman nonparametric test ranking of KUMA and all benchmarks. All results are significant.

| Model | MSE | | | | MAE | | | |
|---|---|---|---|---|---|---|---|---|
| | Rank | $z$-value | unadjusted $p$ | $p_{\text{Bonf}}$ | Rank | $z$-value | unadjusted $p$ | $p_{\text{Bonf}}$ |
| Koopa | 5.10 | 3.022663 | 0.002506 | 0.037584 | 5.46 | 3.408534 | 0.000653 | 0.009797 |
| U-Mixer | 7.85 | 5.852389 | 4.85e-09 | 7.27e-08 | 6.83 | 4.823398 | 1.41e-06 | 2.12e-05 |
| WaveRoRA | 6.90 | 4.866272 | 1.14e-06 | 1.71e-05 | 6.68 | 4.662618 | 3.12e-06 | 4.68e-05 |
| TimeMixer | 6.24 | 4.190997 | 2.78e-05 | 0.000417 | 10.10 | 8.189057 | 2.22e-16 | 3.33e-15 |
| SparseTSF | 9.11 | 7.149347 | 8.72e-13 | 1.31e-11 | 7.27 | 5.273581 | 1.34e-07 | 2.01e-06 |
| S-Mamba | 5.01 | 2.926195 | 0.003431 | 0.05147 | 4.77 | 2.701103 | 0.006911 | 0.103665 |
| iTransformer | 5.69 | 3.622908 | 0.000291 | 0.00437 | 4.83 | 2.765415 | 0.005685 | 0.085276 |
| RLinear | 10.79 | 8.875052 | 0 | 0 | 9.28 | 7.342283 | 2.10e-13 | 3.15e-12 |
| PatchTST | 7.55 | 5.541548 | 3.00e-08 | 4.50e-07 | 6.68 | 4.662618 | 3.12e-06 | 4.68e-05 |
| Crossformer | 11.94 | 10.054105 | 0 | 0 | 12.17 | 10.311352 | 0 | 0 |
| TiDE | 13.66 | 11.822684 | 0 | 0 | 13.94 | 12.133525 | 0 | 0 |
| TimesNet | 8.05 | 6.056044 | 1.40e-09 | 2.09e-08 | 7.86 | 5.884545 | 3.99e-09 | 5.99e-08 |
| DLinear | 10.40 | 8.467743 | 0 | 0 | 11.54 | 9.668233 | 0 | 0 |
| FEDformer | 10.70 | 8.778584 | 0 | 0 | 11.41 | 9.52889 | 0 | 0 |
| Autoformer | 14.84 | 13.044611 | 0 | 0 | 15.03 | 13.258984 | 0 | 0 |
| KUMA | 2.17 | - | - | - | 2.15 | - | - | - |

*Table 12.* The Wilcoxon signed-rank test of KUMA and all benchmarks. Stat: the computed test statistic. $p$-value: the computed probability. All results are significant.

| Model | MAE | | MSE | |
|---|---|---|---|---|
| | Stat | $p$-value $(10^{-10})$ | Stat | $p$-value $(10^{-10})$ |
| Koopa | 75 | 17.640 | 126 | 1406.002 |
| U-Mixer | 106 | 293.040 | 52 | 1.430 |
| WaveRoRA | 153 | 9282.804 | 160 | 14596.065 |
| TimeMixer | 5 | 0.000 | 218 | 387745.657 |
| SparseTSF | 49 | 0.992 | 22 | 0.019 |
| S-Mamba | 23 | 0.023 | 6 | 0.000 |
| iTransformer | 104 | 248.181 | 42 | 0.404 |
| RLinear | 48 | 0.876 | 6 | 0.000 |
| PatchTST | 29 | 0.062 | 29 | 0.062 |
| Crossformer | 0 | 0.000 | 1 | 0.000 |
| TiDE | 0 | 0.000 | 0 | 0.000 |
| TimesNet | 1 | 0.000 | 0 | 0.000 |
| DLinear | 0 | 0.000 | 12 | 0.002 |
| FEDformer | 0 | 0.000 | 6 | 0.000 |
| Autoformer | 0 | 0.000 | 0 | 0.000 |
| KUMA | - | - | - | - |

