# OpenReview forum: "KUMA: A Novel Framework with Koopman Separation and Efficient Multilevel Extraction in Time Series Forecasting"
_ICML.cc/2026/Conference — ICML 2026 regular_

### Official Review · Reviewer_7bG7 · 2026-03-02

**Soundness:** 3
**Presentation:** 3
**Significance:** 3
**Originality:** 3
**Overall Recommendation:** 4
**Confidence:** 4

**Summary:**

This paper proposes KUMA, a novel time series forecasting framework designed to resolve high computational complexity, inefficient token utilization, and temporal distribution shifts (TDS). KUMA introduces an input-dependent Koopman module to decompose series into Koopman dynamics and residual dynamics, effectively mitigating operator mixture and alleviating TDS. To process the residual dynamics, the authors design a U-shaped Multilevel Attention (UMA) module that integrates element-wise attention filtering and linear attention. This multilevel architecture achieves a favorable balance between token redundancy and scarcity while maintaining acceptable computational efficiency. Extensive evaluations across 12 benchmark datasets validate KUMA's superior predictive performance compared to existing state-of-the-art approaches.

**Compliance With Llm Reviewing Policy:**

Affirmed.

**Final Justification:**

We will maintain our score 4.

**Key Questions For Authors:**

1. Regarding the implementation, what exactly do $g$ and $\kappa$ represent? Are they simply the linear tokenization layer and the predictor head, respectively? If so, the authors should justify why this architecture can be interpreted as learning a Koopman operator rather than just a general dynamic weight mixing mechanism. It is suggested providing more formal definitions, mathematical derivations, or visualizations  to substantiate this claim.
2. The manuscript primarily provides indirect evidence for mitigating TDS by demonstrating a reduction in overall error. It is suggested the authors additionally validate this on more standard non-stationary evaluations and report results as a function of shift intensity.
3. The manuscript primarily analyzes the complexity with respect to N and D, while the historical length L is compressed to D. It is suggested to supplement training/inference time and memory usage under longer historical lengths (e.g., L=512/1024), and benchmark against models such as PatchTST and iTransformer.
4. The main results are averaged over 5 random seeds, but the standard deviation and confidence intervals are missing. Additionally, it is recommended that the authors clarify the hyperparameter tuning strategies for each baseline.

**Limitations:**

As mentioned above.

**Strengths And Weaknesses:**

Strengths:
1. The experimental evaluation is extensive, encompassing 12 datasets, multiple prediction horizons, and 12 baseline models. The results presented in the main tables and the appendix consistently substantiate the model's superior overall performance, with a particularly pronounced advantage in high-dimensional and multivariate scenarios. Furthermore, the inclusion of ablation studies (Fig. 5), layer-depth analysis (Table 2), and noise robustness tests (Fig. 6) ensures a rigorous and complete validation chain.
2. The manuscript offers a coherent narrative flow, with Figure 2 and Algorithms 1–3 offering a thorough breakdown of the individual modules. Furthermore, the underlying motivations for tackling the three identified challenges (complexity, token-related issues, and TDS) are well-concentrated and logically justified.
3. The integration of input-conditioned Koopman decomposition and U-shaped multi-layer residual modeling within a unified framework represents a novel conceptual synergy. Furthermore, the combination of EWF and linear attention within a hierarchical architecture serves as a concrete implementation specifically tailored for Time Series Forecasting (TSF).

Weaknesses:
1. The KDM appears to be more of a conceptual analogy than a rigorous theoretical application. The manuscript lacks a clear explanation regarding the concrete implementation of $g$ and $\kappa$, and there is a deficiency in the formal derivation showing how Eq. (4)-(5) correspond to Koopman learning principles.
2. Improvements on LVD datasets are marginal and unstable, as the model only achieves parity with baselines like Koopa and U-Mixer in these cases. The authors should delineate the boundaries of the model's effectiveness more objectively rather than using over-generalized claims of superiority.
3. The overall architecture of UMA bears a strong resemblance to existing U-Net or U-Mixer-like frameworks, while the KDM essentially functions as a conditioned low-rank mixture or attention mechanism. It is suggested that the authors provide a more rigorous comparison with models such as Koopa, Autoformer, and U-Mixer. Specifically, it is necessary to highlight the genuinely novel mechanisms introduced in this work and clarify the gap between the proposed approach and strict Koopman learning.

---

> ### Author Rebuttal · Authors · 2026-03-29
>
> Weakness:\
> W2: We will moderate the statement to avoid any overstatement in our revision.\
> W3: Thanks for this readability concern. U-shaped network is generally used in medical image processing and U-Mixer pioneers to combine U-Net and Linear processor. Our work designs the linear attention based on Mamba, leaving out some redundant components. Moreover, filtering introduction is innovative to U-Mixer and aligns with token inefficiency issues. KDM is inspired by Koopman theory and lights up the dimension from [B, N, D] to [B, N, D, R] and back to [B, N, D]. This is the difference from low-rank mixture or attention mechanism. The difference to a strict Koopman operator is that the value is dynamic. We will add remarks to state this point “Koopman Inspired” and include tables containing differences among Koopa, Autoformer, U-Mixer in revision as well as an analog between KDM and standard Koopman operator to more clearly demonstrate the similarities and gaps.\
> Key Question Response:\
> K1/W1: Thanks for your detailed comments to help us make the manuscript better and appreciation of our work. $g$ is a function to help original signals [B, N, D] reach Koopman space, which is generally higher as [B, N, D, R]. In our work, we use linear projection and re-shape operation to achieve this. $κ$ represents a reverse process, in our work, it is not linear, but the combined effectiveness of Eq. (4) and (5) to reach [B, N, D] again. Given weakness 1 together, we decide to add clear analogy and adopt unified mathematics in appendix of the revision. We will also visualize the original signal, Koopman component, residual component to better demonstrate the Koopman. A more detailed tutorials/introduction is written and will be placed on GitHub to enhance readers’ understanding.\
> K2: In original manuscript, we adopted tracking trajectories in Fig. 3 across different datasets to verify that KUMA can reduce temporal distribution shift, as for each abrupt change and local pattern, KUMA can track well in most cases. We further plan to adopt UMAPs in the appendix to visualize the manifolds with/without Koopman module across different temporal patterns. We will try to find a quantitative function (Silhouette Score) to curve TDS.\
> K3: We agree that, in addition to the complexity analysis with respect to N and D, reporting practical training/inference time and memory usage under longer historical lengths (e.g., L=512/1024) would provide more direct evidence of scalability. In the current submission, inference speed and memory are reported under the main experimental settings, while the analysis in Section.D.2 focuses on theoretical complexity because the historical length is compressed in the model pipeline. We agree, however, that this theoretical analysis is only complementary and does not replace practical long-history benchmarking.\
> We also note that increasing L may require model-specific retuning for a fair comparison, so directly reporting only our model at L=512/1024 without re-benchmarking the baselines would indeed be incomplete. Following the reviewer’s suggestion, we will therefore prioritize adding inference latency and GPU memory usage at L=512/1024 in the revision as much as possible, with PatchTST and iTransformer as primary baselines. We believe these additional results will provide a more complete and fair evaluation of scalability under longer historical contexts.\
> K4: We have added standard deviation and confidence intervals in appendix of revisions. We have made detailed tutorials/.md on the project GitHub and plan to publish it after any acceptance. The code hub has feasible .json files with all hyperparameters to maintain reproducibility. Our lab publishes project code hubs to promote academic community.\
> Additional visualizations trying to solve these concerns are presented in https://anonymous.4open.science/r/KUMA/. We hope our responses can help reduce your concerns and make our work closer to ICML.

---

> > ### Author Rebuttal · Reviewer_7bG7 · 2026-04-03
> >
> > We thank the authors for their response. The authors have addressed our primary concerns regarding the paper, and we will maintain our score.

---

> > > ### Author Response · Authors · 2026-04-04
> > >
> > > We are grateful for your constructive comments, which have significantly strengthened our paper. We appreciate you taking our revisions into account and maintain your score.

---

### Official Review · Reviewer_ft1d · 2026-03-07

**Soundness:** 3
**Presentation:** 3
**Significance:** 2
**Originality:** 2
**Overall Recommendation:** 4
**Confidence:** 4

**Summary:**

The authors propose KUMA, which integrates Koopman separation with a U-shaped Multilevel Attention (UMA) to adress the computational complexity, token redundancy, temporal distribution shifts challenges in time seties forecasting. The proposed KUMA employs Input-Dependent method to separate Koopman dynamics from residual to handle non-stationary shifts, and UMA mechanism to balance token redundancy and scarcity. The experiments and analysis demonstrates better empirical results of KUMA against its variants and selected baselines.

**Compliance With Llm Reviewing Policy:**

Affirmed.

**Final Justification:**

My concerns have been adequately addressed.

**Key Questions For Authors:**

See Weaknesses.

**Limitations:**

yes

**Strengths And Weaknesses:**

## Strengths
* **S1:** The Koopman Dynamic Module effectively separates time series into Koopman and residual dynamics, alleviating the affect of temporal distribution shifts.
* **S2:** The U-shaped Multilevel Attention balances token redundancy and scarcity through multilevel encoder-decoder module, maintaining computational efficiency.
* **S3:** The KUMA model achieves state-of-the-art performance selected benchmarks, demonstrating superior accuracy. The ablation studies validate the contribution of each component, showing performance gains.
## Weaknesses
* **W1:** The three challenges: computational complexity, token redundancy, temporal distribution shifts, are well-known in broader study fields, and the paper lacks sufficient justification for why Koopman separation and UMA are advantageous, when compared to prior approaches.
* **W2:** The [OLinear](https://arxiv.org/pdf/2505.08550) is a closely related method to KUMA, which is based on Pearson matrix orthogonal decomposition for input-dependent separation, is not discussed or compared with KUMA in either theory or experiments.
* **W3:** Most of the selected baselines are before 2024 , which limits the validation of KUMA's effectiveness against more recent and stronger forecasters.

---

> ### Author Rebuttal · Authors · 2026-03-29
>
> K1:\
> Thank you for these comments. We agree that computational complexity, token redundancy, and temporal distribution shifts have been studied individually in broader domains. Our point is not that these issues are newly identified, but that they frequently co-occur and interact in TSF, while many existing methods address only one or two of them, with nontrivial trade-offs. Our motivation is to provide a more unified treatment that balances these challenges within a single framework.\
> Regarding Koopman component, the motivation is to map the input into components with more regular evolution patterns, which makes prediction under temporal shifts easier to handle. As described in the Introduction and illustrated in Fig. 1 and Fig. 9, KMD design is introduced to avoid operators mixtures, which can incur substantial memory overhead. This advantage is reflected in Fig. 4, where KUMA uses 2.39 GB memory versus 5.57 GB for Koopa. In addition, Fig. 3 visualizes the tracking trajectories of KUMA and suggests that the adaptive Koopman scheme improves temporal alignment and stabilizes the evolution process. While amplitude fitting is still imperfect in some hard cases, the figure primarily supports improved temporal alignment and stabilization.\
> Meanwhile, UMA is designed to address the token inefficiency issue. Its multilevel aggregation/filtering mechanism aims to suppress redundant interactions while preserving informative dependencies, thereby offering a better balance between efficiency and representation quality than methods that only compress tokens or only enlarge receptive fields. Table 2 further supports this design by showing how different layer numbers affect different datasets, which is consistent with our motivation that token processing should be adaptive rather than fixed.\
> In the revision, we will strengthen the discussion of how Koopman separation and UMA address different but coupled issues, and we will add more visualizations, such as layer-wise heatmaps and token statistics, in the supplement to better illustrate the design motivation and empirical behavior of the proposed framework.\
> K2:\
> We review OLinear from scratch. OLinear and KUMA are indeed related in that both avoid forecasting directly in the entangled original time domain. However, the two methods are based on different principles and motivations. OLinear performs forecasting in an orthogonally transformed domain, which is not time, frequency, wavelet domain. The core algorithm is based on diagonalizing Pearson correlation computation and use square matrices Q and transposed Q to form decorrelating temporal dependencies and improving prediction in the transformed space. By contrast, KUMA first is not based on Pearson correlation matrix but Koopman theory and always in time domain and Koopman space. It is designed to separate the input into components with more regular evolution patterns, to better handle temporal shifts while maintaining adaptive dynamics modeling. The adaptive scheme is to avoid meta-Koompan operators if there are many patterns at the cost of breaking standard Koopman theory.\
> The two methods also differ at the architecture level. OLinear is built as a linear forecasting model with its NormLin (CSL and ISL) module for multivariate dependency modeling, whereas KUMA employs Koopman module as a component and further introduces UMA to address token inefficiency. Therefore, OLinear is not equivalent to KUMA either theoretically or architecturally.\
> OLinear provides a good direction for us to solve some issues KUMA may face with. As KUMA does not involve coordinate or Transformed domains, therefore, it is better for us to select OLinear as a first-tier baseline when KUMA is extended to Transformed domain.\
> K3:\
> We agree that comparison against more recent and stronger forecasters would make the empirical validation more complete. Our current baselines were selected to cover representative and competitive forecasting families, rather than recency alone and the motivation/selection standard of each baseline is stated in E.2 Section. In addition to widely used strong baselines such as PatchTST and RLinear, we also include several more recent representative methods, including iTransformer, S-Mamba, Koopa, and U-Mixer. Our goal is to compare KUMA against competitive models from different forecasting paradigms (KUMA relates to), rather than selecting baselines purely by publication year.\
> We nevertheless agree that adding even more recent methods would make the evaluation more complete. In the revision, we will further clarify the rationale behind our baseline selection and, where feasible, include additional recent strong baselines in the supplement/revision. Additional visualizations trying to solve these concerns are presented in https://anonymous.4open.science/r/KUMA/. We hope our additional efforts can improve your impressions on our work.

---

> > ### Author Rebuttal · Reviewer_ft1d · 2026-04-02
> >
> > Thank you for your response.
> >
> > Regarding W1, I would like to clarify my point: there are already numerous existing methods in the time series forecasting literature that attempt to address the three challenges you identified. The manuscript currently lacks a discussion of what specific attempts these prior methods have made and how their performance compares to KUMA. I believe it is essential to incorporate such a comparative discussion to better position your contributions.
> >
> > Additionally, my concern in W3 remains unresolved. I still believe that comparing KUMA with more recent and strong baselines is necessary to solidify and demonstrate the effectiveness.

---

> > > ### Author Response · Authors · 2026-04-04
> > >
> > > Thanks for your deeper discussion. We make further responses for your remaining concerns.
> > >
> > > 1. Token inefficiency, computational efficiency, temporal distribution shift, these three challenges are currently explored by many researchers including us. However, our manuscript further attempts to solve another issue: mixture of Koopman operators with increasing computational complexity when faced with increasing patterns. Moreover, for token inefficiency, we observed that Transformer-based methods can tackle complex scenarios well, but in simple cases, there are many redundant tokens negatively affecting the predictive performance. On contrast, while other lightweight models such as Mamba, Mixer variants are good at moderate and simple scenarios, the token density is sparse, triggering token scarcity. Therefore, we introduce U-Net structure to achieve flexible adjustment. To maintain lightweight, we merely keep the core components of Mamba to design linear attention blocks as processors of UMA. For temporal distribution shift, we import Koopman theory. In previous works, the put efforts in designing observation function or mixture of operators to enhance adaptability to different patterns. However, both will increase additional computational complexity for each pattern. Our work is inspired by Koopman theory and adopt only one operator to be adaptive to different patterns. However, the scheme we proposed suffer from abrupt change loss. The is the necessity of introduction of UMA for residual component which contains abrupt change. Therefore, our work has its own scientific values. We also take your concerns into consideration, and in the Related Works section of the revision, we will add detailed discussions on what previous peer works have done and their cons and pros. We hope after these explanations and modifications can reduce your concern on this point.
> > >
> > > 2. For the baseline issues, we add three additional baselines: [1] SparseTSF (2026), [2] WaveRoRA (2025), [3] TimeMixer (2024). The previous baselines include: Koopa (2023), U-Mixer (2024), S-Mamba (2025), RLinear (2023), DLinear (2023), TiDE (2023), TimesNet (2023), iTransformer (2024), PatchTST (2023), Crossformer (2023), FEDformer (2022), Autoformer (2021).
> > >
> > > The updated performance comparison among all baselines is published at https://anonymous.4open.science/r/KUMA/ and we will include them the revision. We hope these additional experiments can better clarify the proper position of KUMA nowadays and reduce your concerns.
> > >
> > > We hope all these responses can allow our work to be closer to ICML and receive fair appreciations.
> > >
> > > [1] Shengsheng Lin, Weiwei Lin, Wentai Wu, Haojun Chen, C. L. Philip Chen, “SparseTSF: Lightweight and Robust Time Series Forecasting via Sparse Modeling," in IEEE Transactions on Pattern Analysis and Machine Intelligence, vol. 48, no. 1, pp. 170-183, Jan. 2026.
> > >
> > > [2] Aobo Liang, Yan Sun, Nadra Guizani, "WaveRoRA: Wavelet Rotary Route Attention for Multivariate Time Series Forecasting," in Transactions on Mobile Computing, vol. 25, no.1 pp. 1287-1301, Aug. 2025.
> > >
> > > [3] Shiyu Wang, Haixu Wu, Xiaoming Shi, Tengge Hu, Huakun Luo, Lintao Ma, James Y. Zhang, Jun Zhou, “TimeMixer: Decomposable Multiscale Mixing for Time Series Forecasting”, in International Conference on Learning Representations (ICLR), May. 2024.

---

### Official Review · Reviewer_t2M2 · 2026-03-13

**Soundness:** 2
**Presentation:** 3
**Significance:** 3
**Originality:** 3
**Overall Recommendation:** 4
**Confidence:** 3

**Summary:**

The work proposes KUMA, a time-series forecasting model that splits the input into two parts: a Koopman-style dynamics to capture the main patterns and a residualpart for local details and abrupt changes. The residual part is handled via a U-shaped multilevel attention module, and the final goal is to improve forecasting while keeping computation efficient. The paper evaluates the method on numerous benchmark datasets and reports promising overall results.

**Compliance With Llm Reviewing Policy:**

Affirmed.

**Final Justification:**

The rebuttal has fully addressed my concerns. I will raise my recommendation score to 4.

**Key Questions For Authors:**

1. The elementwise filtering uses projected features and computes a score from elementwise multiplication, then applies a sigmoid gate and residual form X_{EWF} = Weighting×EV+EV. That may work, but it is not obvious why this is the right mechanism for handling the claimed “Matthew effect,” or why it should capture both temporal dependency and variate correlation in a principled way.

2. The seperation step is simply X_K = KD, X_R = X_{emd} - X_K. There is no orthogonality constraint, no reconstruction check, and no identifiability argument. So there is nothing preventing the Koopman and residual parts from mixing information arbitrarily.

The paper's approach is still interesting to me. Please address my questions and concerns.

**Limitations:**

Yes.

**Strengths And Weaknesses:**

Strengths:
1. Soundness: the motivation and method is fairly clear and makes sense overall. The paper splits the model into a Koopman-style for global patterns and a multi-grained patterns for residual details.

2. Presentation: the main idea is easy to follow.

3. Significance: the problem that the authors try to solve i important, which are long term multivariate forecasting under non-stationary time series.

4. Originality: the paper is novel in combining its pieces. Using a Koopman-inspired together with a U-shaped residual model is a meaningful design choice.

Weaknesses:
1. Soundness:
1.1) Some of the main claims are not fully supported. For example, the paper says the Koopman design helps with temporal distribution shift, but this is not directly proven. The separation step is also very simple, so it is hard to tell, in my opinion, whether it really isolates residual dynamics in a meaningful way or is just a useful heuristic. In the robustness test, and the paper does not report statistical significance.

1.2) The “Koopman” module looks more heuristic than truly Koopman-based. Koopman theory is about find features where time evolution becomes linear. While the method applies a learned nonlinear block and call its output Koopman dynamics.

---

> ### Author Rebuttal · Authors · 2026-03-29
>
> Weakness:\
> Thanks for your insightful comments and appreciation of our work. We agree that the original manuscript has some flaws to better clarify our claims. In the revision, we look through and moderate claims according to the following additional contents:\
> [1] In Fig. 3, we adopted tracking trajectories across different datasets to verify that KUMA can reduce temporal distribution shift, as for each abrupt change and local pattern, KUMA can track well in most cases. We further plan to adopt UMAPs in the appendix to visualize the manifolds with/without Koopman module across different temporal patterns.\
> [2] In fact, our goal is not to claim an exact or uniquely identifiable decomposition, but to introduce a lightweight and efficient separation mechanism that is trainable in an end-to-end way and empirically useful for forecasting. Through visualizations at the beginning of this project, Koopman module can capture most manifolds. We will place such visualizations of the Koopman component, residual component and complete signal to provide a clearer elaboration in the revision for readers.\
> [3] For the statistical testing, we have inserted Friedman nonparametric testing, and Wilcoxon Signed-Rank Testing in the revision. Moreover, a radar chart is imported to better visualize the superiority of KUMA over baselines.\
> [4] Yes. The design of Koopman module is inspired the truly Koopman theory. The key point in classical Koopman is to adopt fixed Koopman operators. However, for different patterns, it will require a large number of operators, and this motivates us to introduce adaptiveness. In our design, the operator value is dynamic to inputs, which breaks the standard principles. Humbly, we will revise the context to clearly reflect the heuristic nature to readers.\
> Key Question Response\
> K1: Our intention was to use “Matthew effect” as an intuitive motivation to describe the tendency that stronger informative signals may be further emphasized while weaker noisy components are suppressed. More precisely, the proposed EWF should be viewed as a lightweight gated residual reweighting mechanism rather than a formal modeling of the Matthew effect. The elementwise interaction produces an adaptive score for each feature position, and the sigmoid gate selectively amplifies or attenuates the corresponding responses while the residual term preserves the base representation. Its role is therefore to improve feature discrimination in a simple and efficient manner. We will add remarks to moderate this claim and avoid presenting EWF as a principled standalone solution to the Matthew effect. Moreover， temporal dependencies and variate correlation have been elaborated in Appendix: Fig. 8.\
> K2: It is admittedly that these two components probably may not fully-independent. The inspiration came from our observation that KUMA-w/o-UMA can capture main patterns/tendency across most cases. The UMA is to solve the drawback of the sole Koopman model. We have added more visualizations, modifications as stated in [2]. KUMA is evolving and orthogonality constraint, reconstruction check will be concluded in the future work.\
> Additional visualizations trying to solve these concerns are presented in https://anonymous.4open.science/r/KUMA/. We hope all these explanations, additional efforts in the revision can address some of your concerns and become more clearer for readers and closer to ICML.

---

> > ### Author Rebuttal · Reviewer_t2M2 · 2026-04-03
> >
> > Thank you for your response. The rebuttal addressed my concerns. I will increase my recommendation to 4.

---

> > > ### Author Response · Authors · 2026-04-04
> > >
> > > We are grateful for your constructive comments, which have significantly strengthened our paper. We appreciate you taking our revisions into account and increasing your score. Sincerely thank you and would you mind please increase your recommendation score at your early convenience  (before the deadline of the rebuttal).

---

### Decision · Program_Chairs · 2026-04-30

**Decision:**

Accept (regular)

**Comment:**

This paper presents a time-series forecasting model that decomposes temporal dynamics into two components: a Koopman-style dynamics and residual dynamics. The effectiveness of the proposed method has been validated through extensive experiments. While reviewers initially raised concerns regarding the connection with the Koopman theory and the incompleteness of comparisons with state-of-the-art time-series forecasting methods, the authors have adequately addressed these issues during the rebuttal period. In summary, all reviewers lean toward accepting the paper.